**Interacting effects of land-use change and natural hazards on rice agriculture in the Mekong and Red River**
**Deltas in Vietnam**
Kai Wan Yuen[1], Tang Thi Hanh[2], Vu Duong Quynh[3], Adam D. Switzer[1], Paul Teng[4] , Janice Ser Huay Lee[1]
[1]Earth Observatory of Singapore, Asian School of the Environment, Nanyang Technological University
Singapore
[2]Faculty of Agronomy, Vietnam National University of Agriculture
[3]Institute for Agricultural Environment, Vietnam
[4]S. Rajaratnam School of International Studies, Nanyang Technological University, Singapore
**Abstract**
Vietnam is a major rice producer and much of the rice grown is concentrated in the Red River Delta (RRD) and
the Mekong River Delta (MRD). While the two deltas are highly productive regions, they are vulnerable to
natural hazards and the effects of human induced environmental change. To show that the processes and
issues affecting food security are reinforcing, interdependent and operating at multiple scales, we used a
systems-thinking approach to represent the major linkages between anthropogenic land-use and natural
hazards and elaborate on how the drivers and environmental processes interact and influence rice growing
area, rice yield and rice quality in the two deltas. On a local scale, demand for aquaculture and alternative
crops, urban expansion, dike development, sand mining and groundwater extraction decrease rice production
in the two deltas. Regionally, upstream dam construction impacts rice production in the two deltas despite
being distally situated.  Separately, the localized natural hazards that have adversely affected rice production
include droughts, floods and typhoons. Outbreaks of pests and diseases are also common. Climate change
induced sea level rise is a global phenomenon that will affect agricultural productivity. Notably, anthropogenic
developments meant to improve agricultural productivity or increase economic growth can create many
unwanted environmental consequences such as an increase in flooding, saltwater intrusion and land
subsidence, which in turn decreases rice production and quality. In addition, natural hazards may amplify the
problems created by human activities. Our meta-analysis highlights the ways in which a systems-thinking
approach can yield more nuanced perspectives to tackle "wicked" and interrelated environmental challenges.
Given that deltas worldwide are globally significant for food production and are highly stressed and degraded,
a systems-thinking approach can be applied to provide a holistic and contextualized overview of the threats
faced in each location.
**Key words:** system dynamics, rice, climate change, food security, Mekong Delta, Red River Delta, Vietnam






## 1. Introduction

A delta is defined as a low-lying sedimentary landform located at the mouths of rivers. The mixing of fresh and saltwater in these sediment-rich land-ocean coastal zones provides fertile land for agricultural activities to support a large number of people. Besides agriculture, resources in deltas have also been tapped for fisheries, navigation, trade, forestry, fossil energy production and manufacturing. Unfortunately, deltas are highly vulnerable to a range of environmental hazards such as typhoons, floods, storm surges, tsunamis, coastal erosion and seasonal inundations (Syvitski and Saito, 2007). In addition, local human activities, land subsidence, water stresses and global sea level rise have exacerbated their environmental vulnerability (Day et al; 2016; Seto, 2011; Tessler et al. 2015). The threats faced by deltas are considered to be "wicked problems" with no easy solutions to counter them (DeFries and Nagendra, 2017).

In this paper, we focus on the Red River Delta (RRD) and the Mekong River Delta (MRD) in Vietnam as these two deltas are highly populated hubs of agricultural production that are highly vulnerable to environmental hazards. We use a systems-thinking approach to illustrate some of the "wicked problems" present in the two deltas in Vietnam and the implications of these anthropogenic and natural hazard drivers on rice agriculture. Although a variety of crops is cultivated in Vietnam, we focus on rice as it is a staple food for the Vietnamese (Nguyen et al., 2019b; USDA, 2012) and it is also a key export crop. In 2019, Vietnam exported US$1.4 billion of rice and was the fourth largest rice exporter in the world contributing 6.6% of the world's total rice exports (Workman, 2020).

Many studies have investigated how Vietnam is affected by natural hazards or anthropogenic land-use change (cf. Howie, 2005; Minderhoud et al., 2018; Nguyen et al., 2019a; Vinh et al., 2014). Several studies go a step further to examine how changes in anthropogenic land-use have affected rice productivity. For example, higher prices and rising demand for aquaculture and non-rice products have incentivized farmers to shift away from rice monoculture to embrace non-rice crops (Hai, 2019; Morton, 2020). In addition, environmental threats such as worsening saltwater intrusion has limited rice production areas and forced many farmers to convert their now-unusable rice fields into shrimp ponds (Kotera et al., 2005; Nguyen et al., 2017) or turn to growing salt tolerant crops such as coconut, mango and sugar cane (Nguyen and Vo, 2017). Urban expansion is also another key factor that has reduced rice growing areas although agricultural intensification has kept rice yields high despite shrinking growing areas (Drebold, 2017; Morton, 2020).

Meanwhile, the construction of high dikes to mitigate flooding in the Mekong Delta has facilitated triple cropping of rice and increased yields. However, these high dikes reduce the availability of fertile silt and force farmers to rely on costly agrochemicals to maintain yields (Chapman and Darby, 2016; Tran and Weger, 2018). Though there is substantial research on sand mining and upstream dam construction, these non-related anthropogenic factors are often not linked to agricultural productivity even though reduced sediment availability and increased channel erosion would have adverse implications on agricultural productivity (Binh et al., 2020; Park et al., 2020; Jordan et al., 2019)

Besides land-use change, rice grown in the RRD and MRD are susceptible to damage from natural hazards such as typhoons, floods and droughts (Chan et al., 2012; 2015; Grosjean et al., 2016; Terry et al., 2012). Rice crops can be damaged by strong winds and flooding from heavy rain associated with a typhoon event. Rice damage is worse if the typhoon occurs during the vulnerable heading or harvesting periods (Masutomi et al., 2012). In addition, floods can also be caused by heavy monsoonal rains. Notably, moderate levels of freshwater flooding may be beneficial to agricultural production (Chapman et al., 2016). On the other hand, droughts while uncommon have caused millions in economic loss, particularly in the agriculture sector (Grosjean et al., 2016). The most recent 2015-2016 drought affected all the Mekong Delta provinces and caused up to US$360 million in damage, of which US$300 million was agriculture and aquaculture-related damage (Nguyen, 2017).

Arias et al. (2019) conceptually integrated local and regional drivers of change to illustrate the factors contributing to environmental change in the Mekong floodplains. Similarly, Nguyen et al. (2019b) recognized

that there are various drivers of change associated with adapting to widespread salinity intrusion in the Mekong and Red River Deltas and these drivers are constantly interacting with and providing feedback to each other. On a more technical level, Chapman and Darby (2016) used system dynamics modelling to simulate the delays, feedbacks and tipping points between farmers' socioeconomic status and the practice of double or triple cropping in An Giang province in the MRD. Building on the framework provided by these studies, we use systems-thinking to present an overarching picture of how anthropogenic and natural hazard drivers can interact to reinforce or diminish rice production. While many studies may have highlighted the links between the various drivers and rice production, we seek to integrate the different drivers of anthropogenic change and natural hazards to show their inter-related and interdependent nature and how the processes associated with each driver affects rice growing areas, rice production and rice quality in general.

The use of systems-thinking is appropriate as the "wicked" environmental problems present in the deltas of Vietnam are caused by a range of interdependent anthropogenic and natural hazards drivers operating at multiple scales with no easily identifiable, predefined solutions. While interventions may be made to ameliorate problems, these interventions may create feedbacks and unanticipated outcomes (Rittel and Webber, 1973; DeFries and Nagendra, 2017). In addition, systems-thinking can be applied in a range of contexts and at multiple scales. Geist and Lambin (2002) and Lim et al. (2017) applied a system-dynamics approach to understand drivers of deforestation and forest degradation at the national and global scales while Ziegler et al. (2016) used a transdisciplinary learning approach to understand the role of environmental and cultural factors in driving the development of human diseases in Northeast Thailand at the landscape scale.

Our aim is to use a literature review to develop flow diagrams to represent the major linkages between anthropogenic land-use factors and natural hazards and elaborate on how they interact and influence rice productivity in these two deltas. Due to the importance of Vietnam as a major rice producer and exporter in Southeast Asia, as well as the range of threats faced by the rice sector from natural hazards and anthropogenic land-use, we hope to show how the processes and issues affecting food security are not one dimensional and linear but in fact reinforcing and interdependent. Lastly, given that deltas worldwide are globally significant for food production and are highly stressed and degraded landscapes, we argue that a systems-thinking approach can be applied to provide a holistic and contextualized overview of the threats faced in each location.

## 2. Methods

### 2.1. Study sites

The Mekong River Delta (MRD) is the world's third largest delta with a physical area of 4 million ha and it is the larger of the two deltas in Vietnam (Schneider and Asch, 2020; Figure 1). In 2018, the planted area for spring, autumn and winter paddies was 1,573.5 thousand ha, 2,336.5 thousand ha and 197.2 thousand ha respectively. In total, 4.1 million ha of rice was planted over the three planting seasons with 24,507 thousand tons of rice produced. The delta is home to 17.8 million people with many dependent on agriculture for their livelihoods. That 54% of Vietnam's rice is grown in the MRD and most of it is exported overseas makes it strategically important for the Vietnamese economy and for global food security (Chapman et al., 2016; Cosslett and Cosslett, 2018; General Statistics Office of Vietnam, 2020). Up north, the Red River Delta (RRD) is the next largest with a physical delta area of 1.5 million ha (Figure 1; Schneider and Asch, 2020). In 2018, 1 million ha of planted rice produced 6,296.1 thousand tons of rice, the equivalent of 14% of Vietnam's total rice production (524.3 and 516.4 thousand ha of rice was planted in the spring and winter seasons respectively). Approximately 21.6 million people live in the RRD with many also dependent on agriculture (General Statistics Office of Vietnam, 2020).

Soils in the MRD are highly variable with alluvial, acid sulphate and saline soils dominant. Most of the rice grows on the highly fertile alluvial soils which are found in only 30% of the delta (GRSP, 2013). Conversely, soils in the RRD consist of Holocene delta sediments. These Holocene delta sediments are relatively fine-grained

muds and sands, up to 30 m thick that are the product of rapid progradation during the Holocene high sea
level stand (Mathers and Zalasiewicz, 1999). The Holocene sequence overlies coarse-grained Pleistocene
sediments dominated by braided river and alluvial fan deposits formed during the last glacial low sea level
stand. The Quaternary sediments are underlain by a >400 m thick layer of Neogene sedimentary rocks that are
made up of conglomerate sandstone, clay and siltstone (Berg et al., 2007).
Climatically, the MRD has a tropical monsoon climate with two distinct seasons, a dry season from December
to April and a rainy season from May to November. It is generally warm year round with average temperatures
in 2018 ranging from 26°C in December, January and Feb to 29°C in May. The annual rainfall is between 2,000
to 2,400 mm (General Statistics Office, 2018; Kotera et al., 2008). Conversely, the RRD has a tropical monsoon
climate with three seasons: (1) a hot and wet season from May to September, (2) a cool and dry season from
October to January and (3) a cool and humid season from February to April. The hot and wet season is
characterized by high temperatures and high rainfall, the cool and dry season has moderate to low
temperatures and low rainfall while the cool and humid season has a low to moderate temperatures and low
rainfall (Huong et al., 2013; Li et al., 2006). In 2018, the monthly average temperatures in the RRD ranged from
17°C in Feb to 30°C in June. Average annual rainfall is between 1,300 to 1,800 mm (Li et al., 2006; General
Statistics Office, 2018). Both deltas are low-lying with elevations ranging from 0.7 to 1.2 m above sea level
(Binh et al., 2017).
In the MRD, favorable environmental conditions with ample rainfall, tropical temperatures and fertile alluvial
soils, coupled with an extensive dike and irrigation system, have facilitated the production of three rice crops
annually: winter-spring, summer-autumn and autumn-winter (Table 1; Figure 2). In 2018, the summer-autumn
crop was the largest (12,763.7 thousand tons), the winter-spring crop was the second largest (10,833.7
thousand tons), followed by the autumn-winter crop (909.6 thousand tons) (General Statistics Office of
Vietnam, 2020). Compared to the MRD, rice is planted bi-annually in the RRD, first, from February to June
(spring crop) and a second time from July to October (autumn crop) (Table 1; Figure 3). The chilly winters
preclude the cultivation of a third crop of rice. Approximately 3,507 thousand tons of rice were produced
during the spring cropping season while 2,789.1 thousand tons were produced during the autumn season in
2018 (General Statistics Office of Vietnam, 2020).
**2.2. Literature review and causal loop diagrams**
We conducted an online search on Scopus, Web of Science, Google, Google Scholar and individual
journal databases to find articles related to the effects of anthropogenic land-use change and natural hazards
on rice agricultural systems in the RRD and/or MRD. Articles related to the environmental impacts of these
anthropogenic interventions and natural hazards were included as these tend to explain the environmental
processes in detail. A range of literature sources including peer-reviewed journal articles, book chapters and
scientific reports from non-governmental organizations were included. In addition, we reviewed the
bibliographies of our articles to follow up with any other relevant literature that was not listed in our search.
Since sea level rise would affect the viability of the two deltas as major rice producing regions (Mainuddin et
al., 2006), we also included relevant articles on sea level rise.
We obtained 126 articles through our literature search (cf. supplementary materials). Every article was
considered to be a single case study and was read in detail by the lead author. Thereafter, the natural or
anthropogenic drivers and/or the environmental process that would lead to a change in rice productivity
directly or indirectly were identified. Adopting a systems-thinking approach, we constructed flow diagrams to
identify and visualize the interconnections among the drivers of rice productivity in both deltas.
We first developed causal links which describe how an anthropogenic driver would influence rice productivity
either directly or through an environmental process. We also documented if each driver had an increasing or
decreasing effect on an environmental process that could influence rice productivity by affecting rice growing
area, rice yield or rice quality. This relationship is represented by an arrow which indicates the direction of

influence, from cause to effect. The polarity of the arrows (plus or minus) indicates whether the effect is increasing or decreasing (Lim et al., 2017). A plus sign indicates that a link has "positive polarity" and a minus sign indicates "negative polarity." The polarity of the causal link between A and B is said to be positive when an increase/decrease in A causes B to increase/decrease. A causal link is negative when an increase/decrease in A causes B to decrease/increase (Newell and Watson, 2002). We constructed two flow diagrams - the first flow diagram describes how anthropogenic land-use drivers affect rice growing area, rice yield per hectare and rice quality in the MRD and RRD (Figure 4), while the second causal flow diagram describes how natural hazards in the MRD and RRD affect rice growing area, rice yield per hectare and rice quality (Figure 6). The references we used are found in the Supplementary Materials.

## 3. Results

### 3.1. Local anthropogenic drivers

### 3.1.1. Aquaculture and alternative crops

Both deltas face widespread salinity intrusion that threatens rice production. In the Mekong Delta, salinity intrusion is a naturally occurring phenomenon during the dry season. Tides from the South China Sea and the Gulf of Thailand bring saltwater inland and salinity intrudes up to 70-90 km inland as the length of sea dikes is limited. There are 1,500 km of sea and estuary dikes in RRD versus 450 km of sea dikes in the MRD (Le et al., 2018; Preston et al., 2003; Pilarcyzk and Nguyen, 2005). Thus, salinity intrusion extends up to 20 km from the main river in the RRD (Ca et al., 1994). As rice plants are unable to thrive in soils with soil salinities exceeding 4 g/L (Pham et al., 2018b), affected farmers have converted their paddy fields into aquaculture ponds to cultivate shrimp and fish instead. Other farmers have turned to planting salinity tolerant crops such as coconut, mango and sugarcane. In some cases, farmers have opted for a rice-aquaculture system whereby rice is planted in the wet season and fish/shrimp is cultivated in the dry season when soil salinities are high (Nguyen and Vo, 2017; Pham et al., 2017).

Besides environmental factors, the greater profitability of fruits, vegetables, fish and shrimp also incentivise farmers to plant more non-rice crops. The income per hectare of rice was USD 146 compared to pomelo (USD 16,844) and coconut (USD 1,484) (Hoang and Tran, 2019). Meanwhile, farmers who practise shrimp-rice rotational systems earned 50% more than those who had two rice crops (Morton, 2020; Schneider and Asch, 2020). In addition, high demand for fruits and vegetables for export and from growing urban populations with greater affluence and knowledge about nutrition, also encouraged farmers to diversify from rice monoculture leading to possible declines in the overall rice growing area (Hai, 2019; Nguyen and Vo, 2017; Figure 4).

Finally, government policies encouraging farmers to move away from growing rice also contributed to the planting of more non rice crops on paddy land. As a result, there is increased use of paddy land for non-rice crops, orchards, freshwater and brackish aquaculture (Van Kien et al., 2020). Using remote sensing to assess land use and land cover change in the Mekong Delta, Liu et al. (2020) found that aquaculture had become the second largest land use type following planted land. This was facilitated by government regulations, salt intrusion and higher profitability of aquaculture productions. In any case, even though farmers may have moved on from growing rice, many still continue to apply excessive amounts of pesticides on their crops (Normile, 2013; Figure 4)

### 3.1.2. Urban expansion

Rapid urbanization is accelerating the loss of agricultural land in both the Red River Delta and Mekong River Delta. In Hanoi, a major city in the Red River Delta, 1,420 ha of agricultural land was lost per year from 2000-2007, equivalent to a yearly loss of 3%. By 2025, up to 450,000 ha of agricultural land are expected to be converted to urban land. Most of this land use conversion occurs in peri-urban areas 5-15 km from the city centre. The same peri-urban land is often used to grow food, flowers and livestock to supply food for the

urban population in Hanoi. However, this land is considered by local authorities to be land reserve for urban
planning, instead of resources for food supply (Drebold, 2017; Pham et al., 2015). Most of this peri-urban
agricultural land is often forcibly obtained with minimal compensation given to farmers and then sold to
foreign developers (Drebold, 2017). Similar urban expansion is also occurring in Can Tho city in the Mekong
Delta with corresponding losses of agricultural land (cf. Garschagen et al., 2011; Pham et al., 2010). A decline
in agricultural land means a decline in rice growing areas as well (Figure 4).
High land prices and a shrinking availability of arable land have forced farmers to practise agricultural
intensification. In the RRD, rice production grew more than 25% from 2000-2011 without corresponding
increases in rice growing areas due to intensified cropping practices involving the use of new high yielding rice
varieties, irrigation during dry season and high inputs of agrochemicals (Drebold, 2017; Morton, 2020). The
high pesticide use was reflected in a study in Nam Dinh province in the RRD where 8 out of 12 target pesticides
were found in agricultural soils. In this study, frequently detected pesticides include isoprothiolane,
chlorpyrifos and propiconazole and besides polluting the environment, the presence of high concentrations of
pesticide residues also lowers the quality of rice sold for consumption (Braun et al., 2018; Figure 4).
In general, pests such as the brown planthopper are naturally occurring and are not a threat at low densities.
However, intensive rice production with high seeding densities and the use of susceptible varieties creates a
constant supply of food which allows their numbers to balloon. This is exacerbated by the asynchronous
planting which creates a continuous supply of rice plants throughout the year in the Mekong Delta. In addition,
the over-use of nitrogen fertilizers increase the pests' reproductive potential. Thirdly, the excessive use of
pesticides also kills the natural enemies of pests such as spiders, ants, bees, beetles, dragonflies, frogs, lady
bugs and wasps. Besides killing the natural predators of rice pests, the pests targeted by the pesticides may
also become resistant to the pesticide. As a result, higher doses of the pesticide may be needed to kill them in
future. For example, killing plant hoppers now requires a pesticide dose 500 times more than was needed in
the past (Normile, 2013). The overuse of pesticide is due to a combination of factors such as insufficient
knowledge of its proper use as well as aggressive marketing by agrochemical companies (Bottrell and Schoenly,
2012; Normile, 2013; Sebesvari et al., 2011). The hashed lines in Figure 4 show that pesticide use may not
necessarily reduce the incidence of pests and diseases if excessive volumes were applied.
**3.1.3. Dikes**
Wet season flooding is a naturally occurring phenomenon in the two deltas of Vietnam (Chan et al.,
2012; 2015). To facilitate the planting of rice during the wet season, flood prevention dikes were constructed
to keep floodwaters out (Figure 5). The MRD has more than 13,000 km of flood prevention dikes; of which
8,000 km are low dikes below two meters tall. These low dikes were mostly constructed before 2000 to delay
the entry of floodwaters at the start of the monsoon season to allow two rice crops to be grown. A severe
flood in 2000 provided the impetus for river dikes to be heightened to 3.5 m to completely keep floodwaters
out. These high dikes have facilitated triple cropping in the MRD, particularly in Dong Thap and An Giang
provinces (Chapman et al, 2016; Howie, 2005; Le et al., 2018; Triet et al., 2017). However, the presence of high
dikes in the MRD has reduced the supply of fertile alluvium, increasing the need for artificial fertilizers and
pesticides to maintain yields (Chapman et al., 2017; Figure 4).
A study comparing sediment deposition in areas of high and low dikes in An Giang Province found that double
cropping farmers who cultivate their crops in areas with low dikes have an average of 2.5 cm of sediment
deposition. This deposition by floodwaters improved their average annual input efficiency by 0.3 tons of yield
per ton of fertilizer. Conversely, triple cropping farmers had very little deposition, averaging 0.5 cm as the high
dikes kept floodwaters out. Some deposition was found only if there had been a dike breach which also caused
crop damage (Chapman et al., 2016). The value of flood deposits is reiterated by Manh et al. (2015)'s study
which estimated that the annual deposition of sediment bound nutrient can naturally supply over half of the
fertilizers needed for a season of rice crop. The provision of "free" fertilizers by the encroaching flood waters
benefits the less economically endowed farmers who must purchase artificial fertilizers to maintain yields
(Chapman et al., 2017; Kondolf et al., 2018; Figure 4).
However, poorly planned and/or maintained dikes are not only functionally ineffective against floodwaters or
coastal surges, they may become an amplifier of destruction when their presence creates a false sense of
security which results in intensive development of low lying areas (Mai et al., 2009; Tran et al., 2018). In
addition, areas unprotected by dikes may be more vulnerable to flooding as the excess water has to flow
somewhere. Using a GIS-linked numerical model, Le et al. (2007) confirmed that engineering structures in the
MRD increased water levels and flow velocities in rivers and canals. This in turn increased the risk of flooding in
both non-protected areas and protected areas (due to dike failure). Hashed lines were used in Figure 4 to
show that dikes do not necessarily reduce flooding.
Likewise, the RRD is also heavily diked with 3,000 km of river dikes (Figure 5) but unlike the MRD, high dikes
are absent (Pilarcyzk and Nguyen, 2005). Besides river and flood control dikes, there are also sea dikes and
salinization prevention dikes in both deltas to protect the area from salinity intrusion. There are 1,500 km of
sea and estuary dikes in the RRD. In the MRD, there are 1,290 km of salinization prevention dikes and 450 km
of sea dikes (Le et al., 2018; Pilarczyk and Nguyen, 2005; Figure 4).
**3.1.4. Sand mining**
Sand mining is carried out on a large scale in the Mekong (Kondolf et al., 2018). Fueled by demand
from reclamation, export and construction, 55.2 million tons of sediment were extracted from the Mekong
main stem in Laos, Thailand, Cambodia and Vietnam from 2011 to 2012 (Bravard and Gaillot, 2013; Robert,
2017).  A more recent analysis of bathymetric maps and the local refilling processes by Jordan et al. (2019) put
the amount of sand extracted from the Mekong Delta in 2018 at 17.77 $Mm^3$.
Besides removing large quantities of riverbed sediments, sand mining operations have created numerous pits
and pools. These pits and pools which can be up to 45 m deep then become sediment traps, trapping bedload
from upstream reaches and preventing them from travelling downstream and contributing to the continued
presence and growth of the delta. In addition, bed incision also occurs as the water is sediment starved. The
down-cutting of river banks can propagate upstream and downstream from the extraction sites for many
kilometers in turn affecting river ecosystems over a large area (Kondolf et al., 2018). This bank incision results
in land loss which threatens rice growing areas (Figure 4).
Aggressive sand mining also disrupts natural flooding. A recent study of riverine mining on flood frequency in
the Long Xuyen Quadrangle (LXQ) in the Mekong Delta found that flood frequency had dropped by 7.8% from
2005-2015. Water levels at local gauge stations also showed an overall decreasing trend indicating that the
lowering of the riverbed had reduced the frequency of flooding. Disrupted flood regimes result in reduced
volumes of water and sediments for agricultural production. In addition, floodwaters typically deposit fertile
sediments while flushing the pesticides and fertilizers accumulated from intensive agricultural production.
When the flood frequency decreases, the frequency at which farmlands benefit from these natural soil quality
enhancement decreases. Consequently, soil fertility may decrease over time and lead to declines in rice yields
unless artificial fertilizers are added (Park et al., 2020; Figure 4).
While there are several studies on the diffuse, yet insidious nature of sand mining in the Mekong (cf. Bravard
and Gaillot, 2013; Bruiner et al., 2014; Jordan et al., 2019; Kondolf et al., 2018; Park et al., 2020; Robert, 2017;
Schmitt et al., 2017), the extent of sand mining in the Red River Delta is unclear as there is almost no research
on this issue. We did however come across an article in a Vietnamese newspaper about rampant sand mining
in the Red River and how mining operations have caused erosion in nearby villages (Chinh, 2018). Similar to
the situation in the MRD, the authorities have turned a blind eye to this illegal business (Bravard and Gaillot,
2013; Chinh, 2018).

### 3.1.5. Groundwater extraction

Another example of an anthropogenic development creating other interrelated problems is that of groundwater extraction. While groundwater extraction has increased the availability of water for human activities, it has exacerbated land subsidence which has increased the severity and extent of saltwater intrusion and reduced the suitability of land for rice cultivation (Figure 4). Minderhoud et al. (2017) developed a 3D numerical groundwater flow model of the MRD surface and found that subsidence rates from groundwater extraction were between 1.1 and 2.5 cm/year. The model also showed that 25 years of groundwater extraction since 1991 had resulted in a cumulative average of 18 cm of subsidence with some hotspots recording over 30 cm of subsidence. Land subsidence from excessive groundwater extraction acts as a catalyst that increases vulnerability to saltwater intrusion and reduces the availability of land suitable for rice production.

Moreover, rice crops become contaminated with arsenic when arsenic-rich groundwater used for non-agricultural use is discharged into rivers and the river water is used for rice irrigation (Lan and Giao, 2017; Minderhoud et al., 2018). High arsenic concentrations in groundwater seem to be of natural origin. In the Mekong Delta, naturally occurring biochemical and hydrological processes cause As to be released from Fe oxides in rocks and sediments into groundwater reservoirs (Fendorf et al., 2010). In addition, deep groundwater extraction causes interbedded clays to compact and expel water containing dissolved As (Erban et al., 2013). Crop quality is reduced when the arsenic enriched water is deposited on topsoils and absorbed by rice plants during growth (Rahman and Hasegawa, 2011; Figure 4).

Similarly, prevalence of groundwater extraction is also high in the Red River Delta. Approximately 70% of the population living in the RRD access water from Holocene and Pleistocene aquifers (Berg et al., 2007; Winkel et al., 2011). Groundwater in the Red River Delta is also contaminated with high levels of As due to reductive dissolution of As from iron oxyhydroxides in buried sediment (Berg et al., 2007; Luu, 2019). Berg et al. (2007) sampled 196 tubewells randomly over a 700 km$^2$ area in the Red River Delta and the concentrations of As in groundwater ranged from 1 to 3050 µg/L with an average of 159 µg/L. Separately, Winkel et al. (2011) collected 512 water samples from private wells in the Red River floodplain and found As concentrations varying from <0.1 to 810 µg/L with 27% of the samples exceeding the WHO guideline value of 10 µg/L. The high concentrations of As reduce the quality of rice harvested in the RRD if As gets into soils and river waters indirectly through the usage of As enriched groundwater (Figure 4). In a study on As accumulation in white rice from the Red River Region in Vietnam, Phuong et al. (1999) reported As values of between 0.03 to 0.47 µg g$^{-1}$ d. wt with the mean value at 0.21 µg g-1 d. wt. The mean value was higher than the mean value reported for Thai rice (0.14 µg g$^{-1}$ d. wt; range: 0.01 to 0.39 µg g-1) (Meharg et al., 2009).

Lastly, although high As concentrations in groundwater is common in in the RRD (cf. Berg et al., 2007; Luu, 2019; Pham et al., 2018a; Winkel et al., 2011), there is no research on (groundwater induced) land subsidence in the Red River Delta. Thus, the magnitude of land subsidence in the delta is uncertain and should be an area for future research.

### 3.2. Regional anthropogenic drivers

### 3.2.1. Upstream dams

The Mekong River originates in the Tibetean Plateau and flows through China, Myanmar, Laos, Thailand, Cambodia and southern Vietnam. To meet growing demands for electricity, many small and large scale hydropower projects have been commissioned in each country to take advantage of this supposedly green and clean source of energy (Nhan and Cao, 2019; Manh et al., 2015). A total of 241 dams have been completed in the entire Mekong Basin with another 29 under construction. A further 91 is currently being planned. These 361 dams consist of 176 hydropower dams and 185 irrigation dams. Of the 364 dams in the Mekong, 20 are in Vietnamese territory (WLE Mekong CGIAR, 2020a). Conversely, the Red River originates in Yunnan province in China and flows towards northern Vietnam. It is less heavily dammed with a total of 105

dams in China and Vietnam. There are 25 hydropower dams, 3 multi-purpose dams and 9 irrigation dams in the Red River Basin in Vietnam (Vinh et al., 2014; WLE Mekong CGIAR, 2020b). While there are no dams in the Mekong River Delta or the Red River Delta due to its relatively flat elevation, upstream dam development influences downstream regions in many ways with the environmental impacts extending far beyond the dam itself (Kondolf et al., 2014).

Firstly, a substantial amount of coarse sand, gravel and suspended sediment is impounded in reservoirs behind the dams instead of being transported downstream. This diminished sediment load may aggravate erosion downstream from the dam (Nhan and Cao, 2019; Figure 4). Using a network model, under a "definitive future" scenario of 38 new dams, the cumulative sediment reduction in the Mekong Delta would be 51%. Conversely, under full build-up of 133 new dams, only 4% of the pre-dam sediment load will reach the Delta (Kondolf et al., 2014). Manh et al. (2015) also reached similar conclusions with a quasi-2D hydrodynamic model of suspended sediment dynamics. Floodplain sedimentation would decrease by about 21 to 96% while sediment load supplied to the sea at the river will diminish by 14 to 95% with the extreme values representing full dam build-up. Even if dam construction was limited to the river tributaries instead of the main stem, the cumulative sediment trapped could be as high as 68% meaning that only about 32% of the sediment load would reach the Mekong River delta (Kondolf et al., 2014).

Indeed, Binh et al. (2020) found that the suspended sediment loads in the MRD had decreased by 74.1% in 2012-2015 primarily due to six mainstream dams in the Lancang cascade in China. In particular, the Manwan and Dachaoshan dams contributed to 32% of the reduction. In addition, from 2014-2017, the average incision rate of the Tien River in the MRD was three times higher than the previously recorded value. Sand mining was responsible for a max of 14.8% of the annual riverbed incision while the remainder was caused by hydropower dams upstream.

The deleterious impacts of dams on sediment loads can also be found in the Red River Delta in spite of the smaller number of dams. While the Hoa Binh dam is located on a tributary of the Red River in Vietnam, its large size has influenced suspended sediment distribution in the lower Red River Basin. For example, an analysis of the suspended sediment concentration over a 50 year period from 1960 to 2010 showed that yearly suspended sediment flux had dropped by 61% at Son Tay near Hanoi (Vinh et al., 2014). Similarly, Duc et al. (2012) calculated that the suspended sediment budget at Son Tay and Hanoi Hydrological monitoring stations was reduced by 56% after the Hoa Binh Dam became operational in 1989. The reduction in sediment loads at the Red River Delta would likewise have a similar impact on delta size and rice growing areas.

Besides a change in sediment loads, dams also alter stream discharge and water levels with concurrent effects on water supplies (not shown in Figure 4). When water levels are high during the rainy season, dams can be used to impound the excess water in the reservoir behind. During dry season when water levels are lower, the dams can release water downstream. In doing so, dams increase dry season discharge and decrease wet season discharge. The modification of seasonal water flows is problematic as changes in natural flow patterns, such as higher flows in dry season and lower flows in wet season would affect rice production as rice growing calendars are currently linked to the natural fluctuations of high and low flows (Robert, 2017). Hence, changes in water levels due to anthropogenic interventions may create unfavorable conditions for crop growth if planting calendars remain unchanged. In addition, lowered water levels during dry seasons due to upstream water impoundment can also lead to increased saltwater intrusion and create unfavorable growing conditions for rice farmers.

**3.3. Local natural hazard drivers**

**3.3.1. Drought**
Droughts do not result solely from a lack of rainfall; they can also result from changes in the arrival of rains and the length of the wet season (Adamson and Bird, 2010; Lassa et al., 2016). Vietnam was affected by droughts

in 1997-1998, 2002-2003, 2009-2010 and most recently in 2015-2016. The 2015-2016 drought was the most
severe in 90 years (Grosjean et al., 2016). All thirteen provinces in the Mekong Delta were affected by the 2015
drought. Besides a lack of water for irrigation, the drought caused saltwater to intrude up to 70 km inland.
Cumulatively, the drought and accompanying saltwater intrusion damaged 400,000 ha of rice crops including
50,000 ha of paddy in Kieng Giang and Ca Mau provinces in the MRD (Grosjean et al., 2016; Nguyen, 2017).
Although there is no research on how droughts and salinity intrusion have affected rice quality in Vietnam,
research from elsewhere has shown that water shortages and salt stress induces physiochemical alterations
which affect the rice grains produced (Pandey et al., 2014; Razzaq et al., 2019; Figure 6).
Compared to the MRD, there is not much research or reports on droughts in the Red River Delta. The UNW-
DPC (2014) reported that the RRD experienced droughts from the end of 1998 to April 1999 which affected
86,140 ha of rice. Another drought occurred from January to February 2004 with the water level of the Red
River at the lowest in 40 years. Low water levels were also reported in 2010, however drought conditions and
saltwater intrusion were more severe in the MRD (Overland, 2010). The effect of droughts on rice agriculture –
reduced yields from a lack of water and salinity intrusion would also be similar in the RRD.
**3.3.2. Freshwater flooding**
417       Ranked as the second most severe natural hazard after typhoons, freshwater floods are caused by
overflowing rivers, heavy monsoonal rains or associated with heavy rain from typhoons (Chan et al., 2012;
2015; Hung et al., 2012; McElwee et al., 2017). Theoretically, flooding reduces rice growing areas but it is
simplistic to assume that flooded fields result in immediate loss of rice crops. A study by Kotera et al. (2005) in
the RRD showed that the type of rice, stage of rice growth, lengths and depths of submergence were factors
that influence the survival rates of rice crops. For example, the local variety Moc Tuyen was less resilient to
submergence than the two other genetically improved high yielding varieties. In terms of growth stage, rice
plants at the tillering stage are more likely to succumb to submergence than those at the vegetative stage.
Plants fully submerged for short durations (two days) also had lower chances of survival than those that were
partially submerged in floodwaters for longer durations (up to eight days). As such the effect of flooding on
rice growing areas is variable as other factors that affect crop mortality include the type of rice grown, the
stage of rice growth as well as the depth and length of submergence in floodwaters (Figure 6).
Although severe flooding can disrupt agricultural activities, moderate levels of freshwater flooding bring
benefits to (rural) farmers (EEPSEA, 2011). Floodwaters from rivers improve agricultural productivity by
depositing nutrient rich flood sediments on agricultural soils (Chapman et al., 2016). In addition, floods wash
away contaminants, purify and recharge aquifers, kill pests and mitigate saltwater intrusion (EEPSEA, 2011;
Hoang et al., 2018; Figure 6). Aquatic resources such as fish, crabs and snails also come in with the floodwaters
which local farmers can collect to supplement their incomes. Besides growing rice, it is also possible to
cultivate vegetables, fish, prawns and ducks in the flooded fields (Nguyen and James, 2013). Towards the end
of the flooding season and the start of the rice cultivation season, floodwaters provide the water needed to
start growing rice (Hoa et al., 2008).
While there are investments in flood prevention measures with the construction and upgrading of river dikes,
the construction and dredging of reservoirs and drainage canals as well as the raising of roads and
embankments, recent floods have caused substantial agricultural losses as current measures have proved to
be inadequate (Hoa et al., 2008; Pilarczyk and Nguyen, 2005). In 2018, 39,000 ha of rice in the RRD was
inundated by heavy rains and floods triggered by Typhoon Son Tinh (VNA, 2018a). Similarly, due to a lack of
embankments and/or poor construction and maintenance of existing embankments, more than 2,000 ha of
rice were lost during the annual floods in 2018. An Giang province was the worst affected, losing 1,270 ha of
rice (VNA, 2018b).

### 3.3.3. Typhoons

Typhoons are the most severe natural hazard that affects Vietnam. When a typhoon occurs, affected areas are exposed to strong winds of up to 50 m/s and up to 300 mm of rainfall in a day. As the rainy season in Vietnam coincides with the typhoon season, widespread flooding can be expected from heavy rain and overflowing rivers (CCFSC, 2005; Mai et al., 2009; Nguyen et al., 2019a). Storm surges can also occur when high winds pushing on the ocean's surface is combined with the effect of low pressure in the center of a typhoon (Takagi et al., 2013). Imamura and Van To (1997)'s study of typhoon disasters in Vietnam since 1950 found that half of the 450 typhoons recorded during the study period were accompanied by a storm surge of over 1 m and 11% were over 2.5 m high. In the Red River Delta, Quynh et al. (1998) found that the maximum storm surge is usually between 1 to 1.5 m above mean sea level. In short, rice production will be adversely affected by strong winds and widespread flooding from heavy precipitation and storm surges in the event of a typhoon. Saltwater flooding may reduce the rice growing areas as rice is not adapted to withstand prolonged submergence and/or saline conditions. Additionally, strong winds damage rice plants with both effects contributing to a reduction in rice yields (Figure 6).

Although a stronger typhoon usually bring higher wind speeds, more rainfall, larger waves and higher storm surges (Larson et al., 2014), the quantity of agricultural losses depends on factors such as landfall location(s) and whether the typhoon occurs during the vulnerable heading or harvesting periods (Masutomi et al., 2012). For example, Typhoon Mirinae (2016) which made landfall in Nam Dinh as a tropical storm damaged or submerged 225,216 ha of rice. Meanwhile Typhoon Nesat (2011) which made landfall in Hai Phong with a similar intensity caused only 3,500 ha of rice damage. Conversely, Typhoon Kalmai (2014) which made landfall in Quang Ninh as a slightly stronger category 1 storm caused 20,000 ha of rice damage (Nhân Dân, 2014; United Nations Vietnam, 2016; Viêt Nam News, 2011).

An average of five to six typhoons affects Vietnam between June and November every year (Larson et al., 2014; Nguyen et al., 2007). Typhoon activity shifts from the north to the south as the year progresses. Therefore, peak activity in the north and southern part of Vietnam is in August and November respectively (Imamura and Van To, 1997). We reviewed the Digital Typhoon (2021) database and found 303 typhoons that came within 500 km of Vietnam's coastline from 1995 to 2018. 29 cyclones made their initial landfall in the Red River Delta while only four cyclones made landfall in the Mekong Delta during the study period – one each in 1973, 1996, 1997 and 2006 (Unpublished results). Although the MRD is less prone to typhoons, the two of the most recent typhoons caused significant damage despite each being classified as a tropical storm upon landfall. Typhoon Linda (1997) caused some 349,232 ha of rice to be submerged while Typhoon Durian (2006) damaged 6,978 ha of agricultural land (International Federation of Red Cross and Red Crescent Societies, 2006; UN Department of Humanitarian Affairs, 1997).

Lastly, typhoons may not necessarily be bad all the time. Darby et al. (2016) combined suspended sediment load data from the Mekong River with hydrological model simulations to examine the role of typhoons in transporting suspended sediments and found that one-third (32%) of the suspended sediment reaching the delta is delivered by runoff generated by rainfall associated with typhoons. When a typhoon affects areas upstream, the land receives higher than usual levels of rainfall which may trigger landslides. This sediment can be transferred into rivers and delivered downstream. While the role of tropical typhoons in sediment mobilization is unclear given the lack of research in this area, such findings have important implications for the MRD as sand mining and upstream dams have caused sharp declines in fluvial sediment loads with corresponding impacts on channel incision and flood frequencies (Brunier et al., 2014; Rubin et al., 2015; Park et al., 2020).

### 3.3.4. Pests and diseases

Examples of pests that occur in rice fields of Vietnam include the brown planthopper (BPH, *Nilaparvata Lugens* Stål), white backed planthopper (WBPH, *Sogatella furcifera* Horvath) and small brown

planthopper (SBPH, *Laodelphax striatellus* Fallen). These planthoppers not only damage plants by ingesting its sap, they also transmit pathogenic viruses that kill the plants. The BPH is a vector for the rice grassy stunt virus (RGSV) and the rice ragged stunt virus (RRSV); the WBH transmits the southern rice black streak dwarf virus while the SBH vectors the rice stripe virus (RSV) and the rice black streaked dwarf virus (RBSDV) (Bottrell and Schoenly, 2012; Matsukawa-Nakata et al., 2019). Other pests that affect rice include parasitic worms called nematodes. Root nematodes that affect deepwater and irrigated rice fields in the MRD include *Hirschmanniella oryzae* (rice root nematode), *Hirschmanniella murcronata* and *Meloidogyne graminicola* (rice root knot nematode). Stem nematodes like the *Ditylenchus angustus* infect floating, deepwater and rainfed lowland rice in the MRD (Nguyen and Prot, 1995). Other significant pests of rice in Vietnam include rice leaf folders, rice thrips and stem borers (Sebesvari et al., 2011). Meanwhile, common rice diseases include bacterial leaf blight, bakanae, black rot of grain, brown spot, leaf yellowing disease, neck blast, rice blast disease, root rot, sheath blight, sheath rot and stem rot (Pinnschmid et al., 1995; Sebesvari et al., 2011; Trung et al., 1995; Kim et al., 1995).

Between 2005 and 2008, rice production in the Mekong Delta was severely reduced by outbreaks of brown planthopper and the associated virulent diseases. The problem was particularly severe in An Giang, Dong Thap and Tien Giang provinces with more than 50% of the cultivated areas affected (Berg and Tam, 2012). In 2009, the Southern rice black streaked dwarf virus affected 19 provinces in North Vietnam including those in the Red River Delta. More than 80% of the rice fields in Nam Dinh, Nghe An, Quang Ninh and Thai Binh provinces were infested and yield was non-existent (Hoang et al., 2011). In short, an increase in pest and/or disease outbreaks will likely cause a reduction in rice yield (Figure 6).

**3.4. Global natural hazard driver**

**3.4.1. Sea level rise (SLR)**

Besides creating new environmental challenges, pre-existing threats to rice production and food security will be exacerbated by climate change. One of the effects of climate change includes rising sea levels. Globally, The IPCC has projected sea levels to rise from a rate of 3.2 mm/year from 1993 to 2010 to as much as 10 mm/year or more by 2010 (Church et al., 2013). This may result in a 0.98 m increase in sea level by 2100 (Lassa et al., 2016). To quantify sea level rise locally, observations at tide gauges across Vietnam have recorded an average yearly increase of 3.3 mm from 1993-2014 (Hens et al., 2018). SLR may also increase the risk of storm surges (Hanh and Furukawa, 2007). In the Red River Delta, Neumann et al. (2015) found that sea level rise through 2050 could reduce the recurrence interval of the current 100 year storm surge with a 5 m height to once every 49 years. Inadequately constructed and poorly maintained dikes and embankments may be breached resulting in saltwater flooding which will damage rice growing areas and other properties (Hanh and Furukawa, 2007; Figure 6).

Rising sea levels coupled with accelerated coastal subsidence caused by excessive groundwater extraction will cause large portions of the low lying RRD and MRD to be inundated and flooded (Allison et al., 2017). This facilitates the infiltration of saltwater into groundwater aquifers and this may increase salinity gradients in the MRD and RRD. In particular, salinity intrusion will worsen during the dry season. Approximately 1.8 million ha in the MRD are already affected by dry season salinity of which 1.3 million ha are affected by salinity levels above 5 g/L (Lassa et al., 2016). This area is predicted to increase to 2.2 million ha with rising sea levels. Meanwhile, in the northeast part of the RRD, the 1% salinity contour has migrated landwards by 4 to 10 km (Hanh and Furukawa, 2007). Increased soil salinization leads to a loss of land available for rice production (Figure 6). Though there are already sea dikes and saline water intrusion sluices in both the MRD and RRD to reduce incursions of seawater (Braun et al., 2018; Tuong et al, 2003), they may be inadequate if they are not well maintained and upgraded.

## 4. Discussion

### 4.1. Untangling complexity

Relevant information on the different drivers and environmental processes affecting rice production in Vietnam are fragmented in a range of academic and non-academic sources (Bosch et al., 2007) making it difficult for policymakers and managers to have a good overview of the reinforcing and interdependent processes and issues affecting food security in Vietnam. Using a systems-thinking approach, we amalgamated the various drivers and created flow diagrams to consider how rice productivity can be positively or negatively impacted by the various drivers and environmental processes (Figures 4, 6). Rice growing areas are negatively affected by the expansion of aquaculture and alternative crops and urban expansion. But with agricultural intensification facilitated by high agrochemical inputs, it was possible to maximize rice yields in spite of a smaller growing area. However excessive agrochemical use affects rice quality and may have an opposite effect on the prevalence of pests and diseases. Next, anthropogenic developments meant to improve agricultural productivity or increase economic growth can create many unwanted environmental consequences. Dikes keep floodwaters and salt intrusion out but there is a reduction in fertile silt deposits. Sand mining increases channel erosion and reduce the frequency of natural freshwater flooding. Similar to sand mining, upstream dams affect sediment accumulation as sediments are trapped in reservoirs upstream but the impacts of upstream dams extend over a larger geographical area. Lastly, groundwater extraction causes land subsidence, saltwater intrusion and arsenic contamination with negative feedbacks on rice growing areas and the quality of rice produced.

Natural hazards not only affect rice quality and quantity but may also amplify some the problems created by human activities – for instance, typhoons and sea level rise may induce saltwater flooding and aggravate salinity intrusion. Conversely, droughts also worsen the extent of salinity intrusion but due to a lack of fresh water. Overall, a substantial reduction in sediment from sand mining and upstream dams, coupled with the process of land subsidence from groundwater extraction and rising sea levels will potentially reduce rice growing areas in future. Besides sea level rise, climate change may also exacerbate the effects of natural hazards by increasing the frequency and severity of natural disasters (cf. Hausfather et al., 2017; Grosjean et al., 2016; Terry et al., 2012). As such, the problems such as excessive saltwater flooding and saltwater intrusion may intensify.

The use of flow diagrams provides a visual overview of the key anthropogenic drivers and natural hazards that affect rice production but we caution that Red River Delta and the Mekong River Delta are vast and diverse regions and there are differences in the ways each delta is affected by natural hazards and anthropogenic drivers. For example, high dikes and the associated problem of sediment exclusion are a problem unique to the Mekong Delta (Chapman et al., 2017). Next, compared to the Mekong, the Red River has substantially fewer dams (361 vs 105). In addition, typhoons are less common in the Mekong Delta and droughts occur less frequently in the Red River Delta.

Within each delta, typhoons tend to affect coastal provinces more than those further inland. Similarly, arsenic contamination and saltwater intrusion is not an issue everywhere across the two deltas. A comparison study of arsenic pollution in the Mekong and Red River Deltas showed that groundwater arsenic concentrations ranged from 1-845 µg/L in the MRD and from 1-3050 µg/L in the RRD. Hotspots with high arsenic concentrations were likely due to local geogenic conditions (Berg et al., 2007). For salinity intrusion, Kotera et al. (2005) measured salinity concentrations in river and canal water across four Mekong Delta provinces and showed that the salinity levels ranged from 0.6 to 14.4 g/L while a localized study in the Nam Dinh province in the RRD showed that salt concentration in river water was higher at the river mouth than in upstream locations. Hence, given the possibility of spatial variations within a large landscape, it is important for local conditions to be taken into consideration.

One limitation of our study is that it was not possible to include all the problems that can potentially affect rice
cultivation in our flow diagrams. We acknowledge issues related to industrial pollution, which may reduce rice
quality and rice productivity (Khai and Yabe, 2012; 2013; Huong et al., 2008). In spite of this, our study
presents the major issues that are common in both deltas and describes how the issues and processes
affecting rice production are interrelated and may operate at different scales. Additionally, a systems-thinking
approach has allowed the multitude of drivers and environmental processes affecting rice production to be
visualized and mapped in a manner that is easy to understand. As ameliorating problems require policymakers
and managers to have a good grasp of the different factors and processes present, a method that considers all
the different drivers and possible unintended consequences from the outset can avoid oversimplifying a
problem and assuming a straightforward solution can be found (DeFries and Nagendra, 2007). For example, to
solve the problem of a shrinking delta, the effects of (high) dikes, sand mining, upstream dams and
groundwater extraction have to be considered. While typhoons may provide some fluvial sediment to offset a
shrinking delta (Darby et al., 2016), the sediment load provided may not be sufficient to offset sediment loss
from sand mining and upstream dams.
**4.2. Adaptation and soft solutions**
Recognizing the environmental challenges limiting agricultural production, farmers in both deltas
have adapted and improvised. Instead of accepting their fate, farmers overcome high soil salinities by
implementing measures such as replacing rice with salinity tolerant crops, transiting to shrimp aquaculture, or
turning to rice-shrimp farming whereby rice is grown in the wet season and shrimp is cultivated in the dry
season. For those unable to switch from rice monoculture, farmers have sought to grow rice on higher ground,
shift crop calendars or dig additional ditches to drain saltwater and store freshwater. In addition, to prevent
water storage ponds from becoming contaminated with saltwater, canvas sheets are placed on the soil surface
to create a protective barrier (Tran et al., 2019; Nguyen et al., 2012). Similarly, farmers in peri-urban areas who
are faced with shrinking agricultural lands have turned to practicing agricultural intensification and/or
switched to planting high value crops such as fruits and vegetables (Morton, 2020; van den Berg et al., 2003).
In short, farmers are constantly experimenting, learning and sharing knowledge and experiences with other
farmers to come up with solutions for overcoming environmental limitations (Tran et al., 2019; Nguyen et al.,
608 2012).

Unfortunately, adaptations are not successful all the time. For example, the widespread conversion of paddy
fields to shrimp ponds will increase soil salinity and reduce the availability of freshwater. Over time, neighbors
who have not switched to aquaculture may be unable to plant rice and would have to seek alternative
livelihoods (Nguyen et al., 2012). In addition, rice-shrimp systems are not problem free as well. Leigh et al.
(2017) found that environmental conditions in rice-shrimp systems were suboptimal and contributed to low
yields and survival. Water temperature and salinity tended to be too high in the dry season and dissolved
oxygen too low, causing most shrimps to die. For rice, the high soil salinity caused by having the aquaculture
pond was a major limitation – in their study, only three out of 18 ponds produced a harvestable rice crop.
Apart from farmer-led initiatives related to crop and land use change, integrated pest management (IPM)
should also be adopted to reduce the use of pesticides to rid pests. Farmers who practise IPM use a
combination of pest resistant cultivars, fertilizer management and agronomic practices to increase the effects
of predators and other naturally occurring biological control agents. For example, farmers can grow flowers,
okra and beans along their paddy fields to attract bees and wasps that infest planthopper pests' eggs. With
more natural predators around, pesticides are only used when necessary (Bottrell and Schoenly, 2012; Normile,
2013). Alternatively, rice-fish farming and duck-rice systems can also be implemented to provide a more
economically and ecologically sustainable alternative to intensive rice monoculture (Berg and Tam, 2012; Men
et al., 2002).
In rice-fish farming, farmers use minimal pesticide as it kills the fish and the natural predators of rice pests.
Instead, fish help to control pests and fish droppings keep the soil fertile. Upon maturity, the fish can be sold
to increase the farmer's income by up to 30% (Berg et al., 2017; Bosma et al., 2012). Ducks can also be reared
in immature rice fields. Besides providing food, the ducks serve as biological controls for insects and weeds.
Their droppings fertilize the soils and their movement aerates the water to benefit the rice plants (Men et al.,
1999; 2002). Men et al. (2002) showed that a duck-rice system in Can Tho province in the Mekong eliminated
the use of pesticides, halved the use of fertilizers and the additional income from the sale of ducks increased
farmers' incomes by 50 to 150%. Overall, the higher incomes and ecosystem services provided by the fish or
ducks, coupled with reduced agrochemical use benefit farmers.
Increasingly, there are calls to move away from three to two rice crops a year in the MRD. Instead of planting a
third crop, floodwaters are allowed to enter the fields to replenish soil nutrients, wash away contaminants, kill
pests and mitigate salinity intrusion. Fish, crabs and snails that arrive with the floodwaters can be collected for
additional income. Triple cropping of rice provides only a single ecosystem service which is marketable rice. On
the other hand, the integration of rice cropping with natural flooding creates a series of positive feedback
mechanisms and ecosystem services that include providing natural pest control and facilitating nutrient cycling
(Nikula, 2018; Tong, 2017).  However, a study by Tran and Weger (2018) in An Giang province revealed that
despite official encouragement to move away from triple cropping, most farmers have largely ignored the
directive as they preferred to earn money from the additional rice crop. In addition, many of them felt that the
benefits of flooding the land were minimal as upstream dams have drastically reduced fertile sediment and
fish. Evidently, farmers are willing to make changes to their farming practices only if it benefits them.
To mitigate "wicked" environmental challenges, there is a need for holistic land use planning and soft
measures (eg. implementing crop and land use change) on top of hard engineering structures. Previously,
management options to increase agricultural productivity and mitigate the threats posed by natural hazards
were largely characterized by hard options such as the construction of dikes, sea walls and sluice gates. These
were typically top-down projects spearheaded by the local government (Neumann et al., 2015; Smajgl et al.,
2015). While highly visible engineering structures are easily constructed and generally effective, unwanted side
effects may be created. For example, flooding and sediment exclusion were some problems that were
inadvertently created due to the presence of high dikes. In the long term, (costly) maintenance is needed to
maintain the functionality of engineered structures (Hoang et al., 2018; Neumann et al., 2015). In addition,
during the pre-construction phase, natural vegetation may be cleared (Geist and Lambin, 2002). Adopting a
systems-thinking approach would allow policymakers and managers to situate the range of mitigation
measures within broader environmental processes. In doing so, a clearer view of the possibilities and
challenges present in an era of widespread anthropogenic development and changing climates is provided.
**5. Conclusions**
The focus of this paper is on the impacts of land use patterns and natural hazards on rice agriculture
in the Mekong and Red River Deltas in Vietnam. While we focused on rice agriculture, these two deltas, like
many other deltas worldwide, are also major production hubs for fruits and vegetables (Day et al., 2016; Nhan
and Cao, 2019). Hence, the natural hazards and anthropogenic factors listed will have an effect on other
agricultural produce as well.
A key finding is that demand for aquaculture and alternative crops and urban expansion has diminished rice
growing areas. The problem of shrinking agricultural land is ameliorated by agricultural intensification which
has increased land efficiency. However, widespread agrochemical use causes land and water pollution and
reduces crop quality. In addition, anthropogenic developments such as dike construction can improve
agricultural productivity but also create unintended environmental problems. Even human activities that are
unrelated to agriculture such as sand mining, groundwater extraction and dam construction can reduce rice
productivity. In addition, natural hazards not only affect rice quality and quantity but may also amplify some
of the problems created by human activities – for instance, typhoons and sea level rise may induce saltwater
flooding and worsen salinity intrusion. In the future, climate change may exacerbate the effects of natural
hazards by increasing the frequency and severity of natural disasters. Therefore, the problems associated with
some of the natural hazards such as excessive saltwater flooding and saltwater intrusion may be more
frequent and possibly worse. In sum, the processes and issues affecting food security are multidimensional and
interdependent and we used a systems-thinking approach to develop a visual representation of the ways in
which anthropogenic land-use factors and natural hazards can affect rice quantity and quality in the MRD and
the RRD in Vietnam. We have also sought to define whether the anthropogenic or natural hazard driver was a
local, regional or global driver to highlight the scale at which each driver operates.
Our review focuses on food security in Vietnam's two deltas but can be applied to other contexts. The
problems present in the two deltas in Vietnam are hardly unique. Across the world, deltas are global food
production hubs that support large populations. Nearly half a billion people live in deltaic regions. Similar to
the Mekong and Red River Delta, large tracts of deltaic wetlands in other countries have been reclaimed for
agriculture, aquaculture, urban and industrial land use. Resultantly, many deltas suffer from flooding,
retreating shorelines due to upstream dams, pollution problems and increasing land subsidence due to
groundwater and mineral extraction. With climate change, rising sea levels will further threaten the viability of
the deltaic landform (Chan et al., 2012; 2015; Day et al., 2016; Giosan et al., 2014; Syvitski et al., 2009).
Given that river deltas worldwide are highly stressed and degraded, a systems-thinking approach can provide a
holistic overview of the "wicked problems" faced in each location and how the various environmental
processes interact with each other. Although our study has focused on rice agriculture in the two deltas in
Vietnam, the application of a systems-thinking approach to evaluate other pertinent phenomena in deltas
elsewhere is a useful tool for understanding how human activities, natural hazards and climate change have
compromised deltaic sustainability.

**6. Data availability**

The papers used in this study are listed in the supplementary materials.

**7. Author contributions**

KW and JSH came up with the idea for this project. KW reviewed papers and wrote the manuscript with
discussions and improvements from all co-authors. JSH, AD, PT, TTH, VDQ provided feedback on the
analysis of data and helped in revisions of the manuscript. JSH provided financial support for this paper.

**8. Competing interests**

The authors declare that they have no conflict of interest.

**9. Acknowledgements**

We thank Khoi Dang Kim, Jose Ma Luis Pangalangan Montesclaros and Nghiem Thi Phuong Le for their advice.
We greatly appreciate the constructive comments and detailed feedback by the two anonymous referees that
helped to improve the quality of this paper.

**10. Financial Support**

This research was supported by the Earth Observatory of Singapore via its funding from the National Research Foundation Singapore and the Singapore Ministry of Education under the Research Centres of Excellence initiative. This work comprises EOS contribution number 360.

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

 **Table**

**Table 1.** Rice planting, growing and harvesting periods in the Mekong River Delta and the Red River Delta in
Vietnam.

| | Planting | | | Harvesting | | | |
|---|---|---|---|---|---|---|---|
| **Mekong River Delta** | **Onset** | **Peak** | **End** | **Onset** | **Peak** | **End** | **Growing period** |
| **Winter-spring** | 1 Nov | 30 Nov | 30 Dec | 15 Feb | 25 Mar | 30 Apr | 115 - 120 days |
| **Summer-autumn** | 15 Mar | 15 Apr | 15 May | 20 Jun | 20 Jul | 25 Aug | 95 - 100 days |
| **Autumn-winter** | 30 Jun | 20 Jul | 20 Aug | 5 Oct | 25 Oct | 30 Nov | 95 - 100 days |
| **Red River Delta** | **Onset** | **Peak** | **End** | **Onset** | **Peak** | **End** | **Growing period** |
| **Spring** | 25 Jan | 10 Feb | 25 Feb | 5 Jun | 15 Jun | 25 Jun | 115 - 130 days |
| **Autumn** | 15 Jun | 1 Jul | 20 Jul | 5 Oct | 25 Oct | 10 Nov | 105 - 110 days |


**Figures**

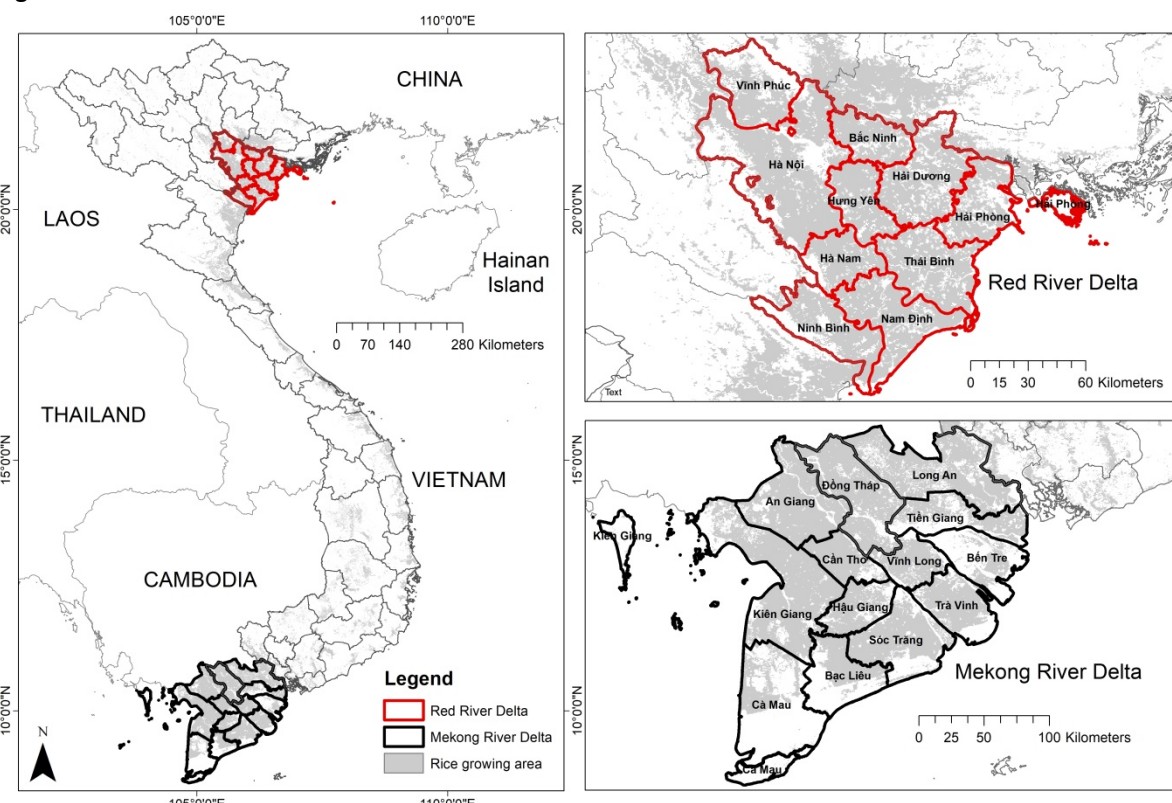


Figure 1. Distribution of rice growing areas in the Red River Delta (RRD) in northern Vietnam and the Mekong
River Delta (MRD) in southern Vietnam. Rice growing extents were obtained from Nelson and Gumma (2015).



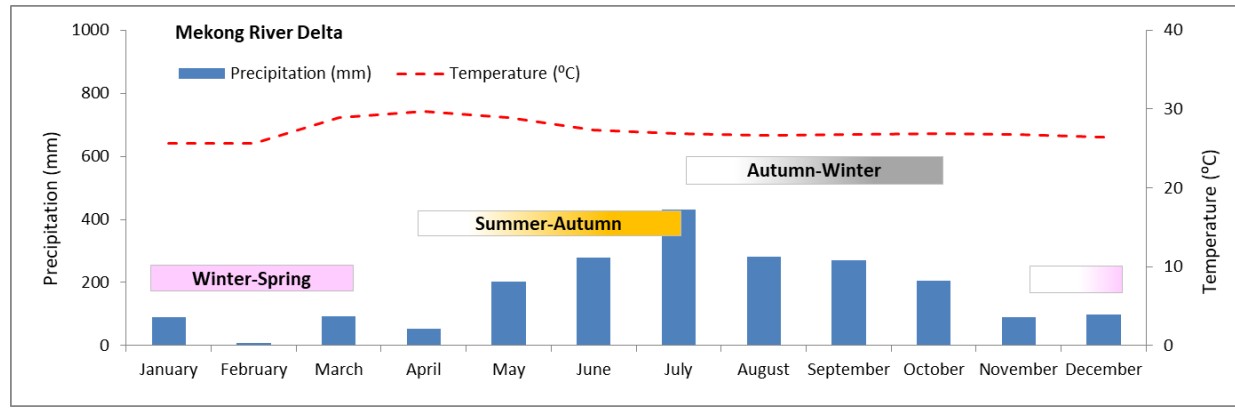


Figure 2. Climograph for the Mekong River Delta annotated with the winter-spring, summer-autumn and
autumn-winter growing seasons. The colour gradient (from faded to dark) represents planting, growing and
harvesting times for each crop. Precipitation and temperature data were taken from the Statistical Yearbook of
Vietnam 2018 (General Statistics Office, 2018).

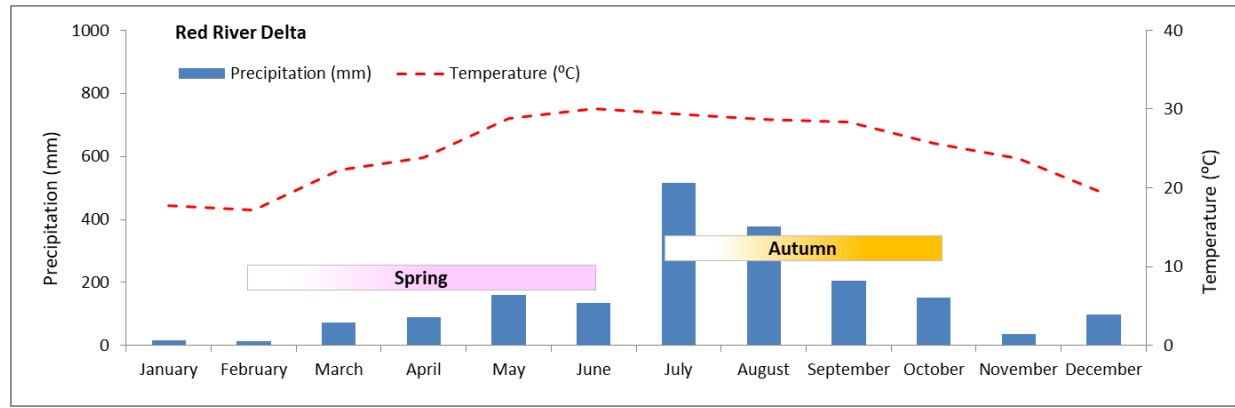


Figure 3. Climograph for the Red River Delta annotated with the spring and autumn growing seasons. The
colour gradient represents planting, growing and harvesting times for each crop. Precipitation and
temperature data were taken from the Statistical Yearbook of Vietnam 2018 (General Statistics Office, 2018).

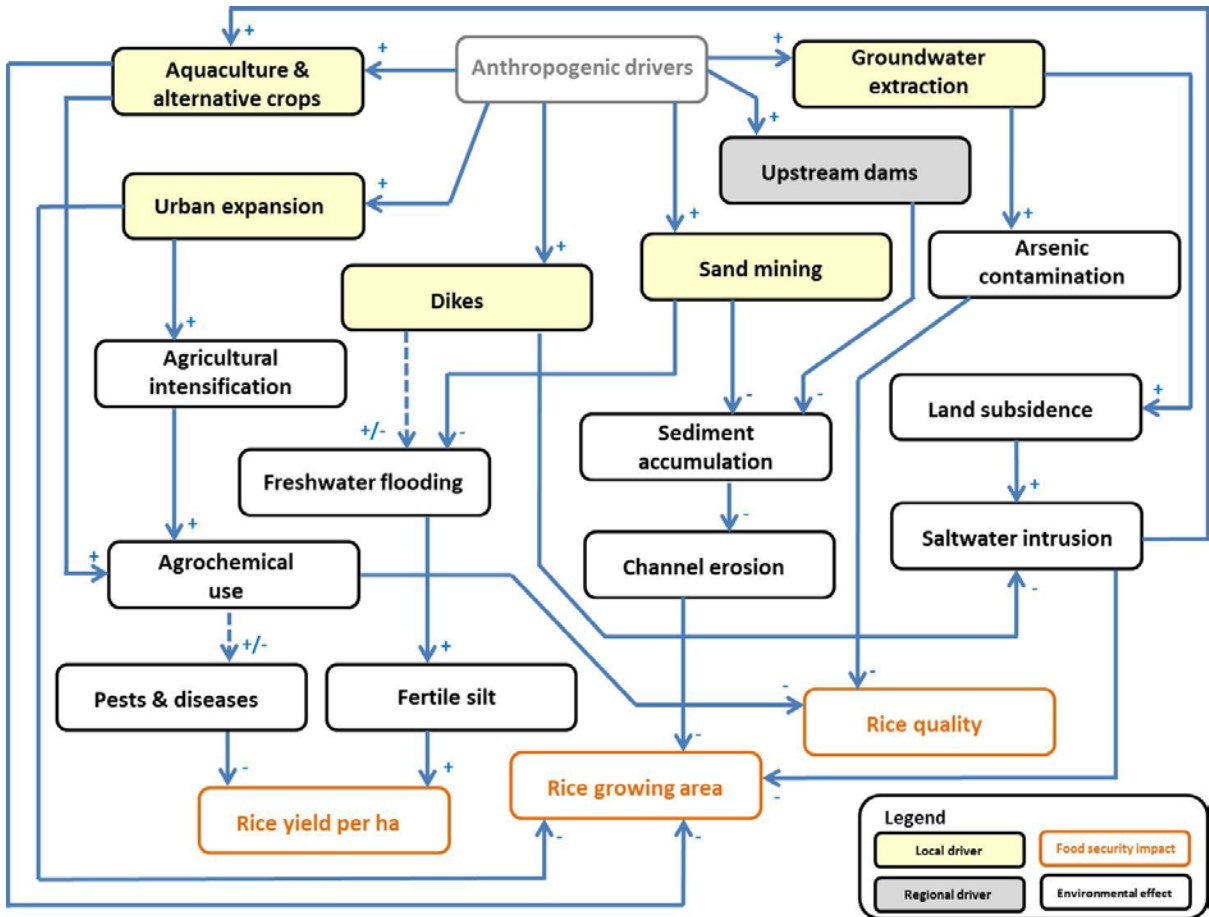

Figure 4. Flow diagram showing the key anthropogenic drivers that affect rice production in the two mega-deltas of Vietnam. The drivers are classified as a local driver if it occurs in the two mega-deltas. Regional drivers are those that occur further away from the two mega-deltas, but within the Asian region. A plus (+) sign indicates that an increase/decrease in A causes B to increase/decrease. A negative (-) sign indicates an increase/decrease in A causes B to decrease/increase. Hashed lines with "+/-" are used when outcomes are unclear. For example, dikes reduce flooding but poorly maintained or planned dikes increase flooding instead. In additions, dikes may potentially cause flooding in unprotected areas. Agrochemical use may reduce the incidence of pests and diseases but the over-use of chemicals can lead to pesticide resistance which may increase outbreaks of pests and diseases.

1328

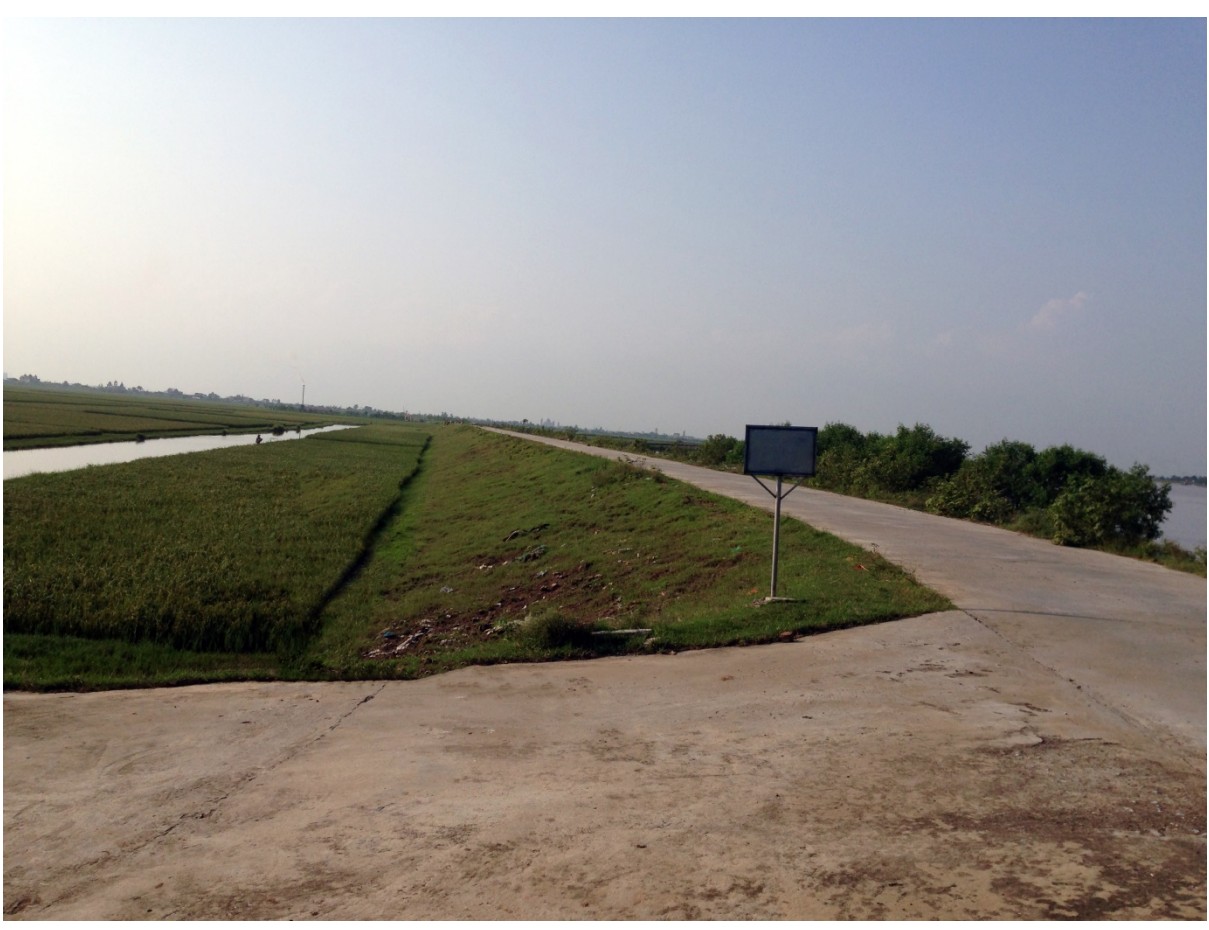

1329     Figure 5. Example of a river dike for flood control in Nam Dinh province in the Red River Delta.

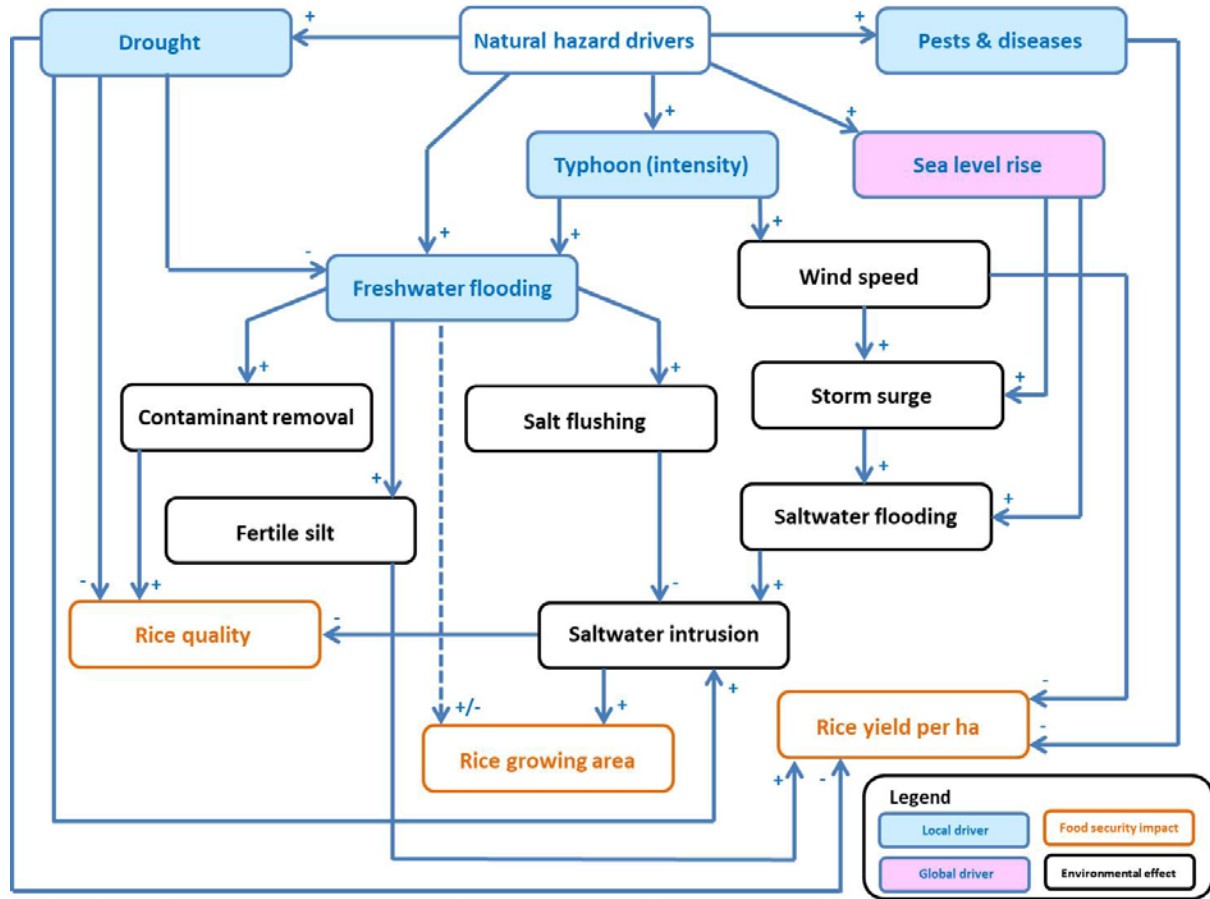

1330

Figure 6. Flow diagram showing the natural hazards that affect rice production in the two mega-deltas of Vietnam.  Local drivers refer to natural hazards that occur within the two mega-deltas. Although sea level rise has implications on a local scale, it is considered a global driver as it occurs on a global scale. The effect of flooding on rice growing areas is variable as other factors that affect crop mortality include the type of rice grown, the stage of rice growth as well as the depth and length of submergence in floodwaters.