# Peer review of "Interacting effects of land-use change and natural hazards on rice agriculture in the Mekong and Red River"

_Natural Hazards and Earth System Sciences, 2020_

## Referee Comment (RC1) · Anonymous Referee #1 · 9 Aug 2020

Dear Dr Gain,

Thank you for the opportunity to review the paper titled "Interacting effects of land-use change, natural hazards and climate change on rice agriculture in Vietnam" (Manuscript Number: nhess-2020-196) by Kai Wan Yuen and colleagues.

The work submitted covers a subject matter that is relevant for NHESS.

The paper certainly has potential and is interesting to read. At present there are several issues that need to be addressed. Most should be relatively straightforward to resolve. If these can be addressed through revisions, then the paper will provide a valuable study on two important deltas in Asia, both of which are vulnerable to natural

degradation and anthropogenic impacts.

My overall general criticisms are:

1. That more information/data/evidence is needed in support of the factors and processes that might impact rice cultivation on the two deltas investigated. At present there is mention of salt intrusion, erosion, sediment extraction, deltaic subsidence, fluvial sediment capture by dams, contamination, loss of soil quality, sea-level rise and increasing frequency/intensity of hazards. All of these are no doubt important for the long term sustainability of the MRD and RRD. But at the moment various statements by the authors about the relative importance of these processes need better grounding. There needs to be inclusion of published data where available, in order to help firm up the main arguments in the discussion. I fully understand that the overarching aim of the paper is to enable visualisation on how all the various influences are connected. But additional data from the literature are nonetheless still needed for proper substantiation.

2. Many minor grammatical errors need correcting throughout the manuscript.

3. Figures 2 and 3 need rethinking and should be improved.

Specific comments and recommendations are listed as below. These are in order as they appear in the manuscript. I trust that the authors will not feel these are unnecessarily critical, but are offered in the spirit of improving the paper for eventual publication.

ABSTRACT

L10. The authors use the term 'mega-deltas'. What does mega-delta mean? This term needs to be defined early on somewhere in the body of the paper.

L12. Remove 'happening'.

L20. Change 'development' to 'growth' to avoid repeating 'development' in the sentence.

L21. Use a comma before 'which' or change to 'that'. This error needs correcting throughout the entire manuscript.

L24. Hyphenate 'systems thinking' because this is used as an adjectival phrase, i.e. say 'a systems-thinking approach'. Do this similarly throughout the manuscript.

INTRODUCTION

L44. Some references are needed here. The following paper and book chapter might be helpful:

'The 'terrific Tongking typhoon' of October 1881 – implications for the Red River Delta (northern Vietnam) in modern times.' Weather, 2012, 67: 72–75. [This gives an example of the severe effect of a typhoon on the rice harvest on the RRD.]

'Impacts of climate change. Challenges of flooding in coastal East Asia.' In: The Routledge Handbook of Environment and Society in Asia. P.G. Harris and G. Lang (Eds), Routledge, Oxford, pp.367–383. Chan, F.K.S., et al. 2015. [Mentions problems of flooding on the RRD and MRD, as well as other deltas in Asia.]

L56. A reference is needed here in support of this statement.

L59. Remove 'related development' (unnecessary text).

L59. Remove 'of'.

L59-62. How can 'coastal dikes ….. lead to a reduction in sediment and water availability…'? This misleading sentence needs rewriting.

L77. Is the mention of 'wicked' problems helpful? What does this even mean? This either needs explaining or omitting.

METHODS

L96. If the area of the MRD is 4 million ha, then how can the area of rice planted on the delta be 4.2 million ha? This is not possible, unless the authors are adding together

the areas of subsequent plantings during the year. Please explain or correct.

L101. Is the RRD 'floodplain' area the same as the delta area? Please give the delta area to be consistent with the description of the MRD above.

L107. Thick 'Quaternary accumulation' of what? Sand, silt or clay? Please briefly give more information on the character of the deltaic sediments.

L109. Remove 'slight'.

L110. Be more careful with grammar. The text should read '. . .the MRD has [not have] . . .. while the RRD has [not have] a temperature of . . .'. This type of error crops up many times throughout the manuscript, e.g. L118, L124, L124. Please ask a native English speaker to check the manuscript carefully for corrections.

L112. Request including climographs for the two deltas, so the reader can more easily understand the annual climatic cycles. Then mark on the graphs the planting, growing and harvesting times of the different rice crops.

L118. Please correct grammar.

L223. Please correct grammar.

L124. Please correct grammar.

L134. Change '. . .a major rice producing region' to '. . .major rice producing regions'.

L151. Change 'describe' to 'describes'.

RESULTS

L193. The authors make a sweeping statement but without supporting data. What is the 'extent of saltwater intrusion'? Is it possible to include a map here to show how saltwater intrusion has been extending into the delta over time?

L196. Where does the arsenic contamination come from? Anthropogenic sources? The authors need to explain otherwise the readers are left guessing.

L203. How much sand mining is occurring? Is it for the construction of dikes? Again, some supporting data are needed.

L204. Again, another sweeping statement about the 'substantial reduction in sediment', but without any supporting data. Please substantiate better.

L204. How much 'land subsidence' has occurred, and over what period? Give rates if available.

With all of the above, if the authors wish to include saltwater intrusion, sand mining, sediment reduction and land subsidence in their Results section, then some additional supporting data are needed.

L225. This statement is wrong. Thermal expansion of seawater does not accelerate the melting of icecaps! Needs rewriting.

L234. The authors mention that the coastline is eroding at a rate of 5 to 10 mm/year. This rate seems far too low. A 0.5 cm rate of coastal retreat per year (0.5 m per century) is insignificant and suggests that the RRD and MRD have nothing to worry about. For comparison, Thailand's Chao Phraya delta front has experienced several km of shoreline retreat over recent decades. Please check the data.

L244. Change text to '...are affected by...' (similar use of plural also needed elsewhere).

L245. What is the local rate of SLR along the coast of Vietnam or in the wider western South China Sea?

L246. Please give some information on the groundwater salinity thresholds for rice cultivation.

L258. Please correct grammar. Look out for similar errors elsewhere through the document.

L266. Please rephrase this sentence.

L272. This contradicts what was said in L118. In other words, what about the irrigation canals mentioned earlier in the paper? Aren't these used for rice irrigation in the absence of sufficient rainfall?

L273. Please correct grammar. Look out for similar errors elsewhere through the document.

L280. What is Cyrtorhinus? An insect, snake, bird, mammal? Please give the English name of this predator.

DISCUSSION

L301. Please correct grammar.

L302. Please correct grammar.

L307. The authors have not provided any information on typhoon and drought frequencies experienced on the two deltas. Please give supporting information earlier in the paper.

L313. '. . .high arsenic concentrations. . .likely due to geogenic conditions'. Please elaborate.

L327. Please correct grammar.

L338. Please correct grammar.

L353. Please correct grammar.

L356. Change 'practice' [noun] to 'practise' [verb].

L364. Please correct grammar.

L371. Please correct grammar.

L386. Please correct grammar.

L386. Please correct grammar (later in the sentence).

CONCLUSIONS

L391. Use ‘Conclusions’.

L411. Please correct grammar.

L414. Needs rephrasing. Do you mean ‘supporting large populations’? [the deltas support large populations – the populations do not support the deltas].

REFERENCES

L521. Reference is in the wrong place.

FIGURES

Figure 1. A cardinal sin has been committed with the maps! Never use a word scale. ‘1 cm = 58 km’ will be incorrect if the published version of the map is not exactly the same size as the original. In fact, the scales must be wrong already as the three maps shown cannot all have the same scale. Use a scale bar instead on each map. These will be correct whatever size the maps are viewed or printed.

Figure 1. Increase the size of the long/lat text. Too small to read.

Figure 1. Label the countries.

Figure 1. L760-764. Unnecessary repetition. The long list of provinces is not needed, as they are already shown on the map.

Figure 2. Strictly speaking, it is incorrect to call this a causal loop diagram as claimed, because ‘rice yield’, ‘rice growing area’ and ‘rice quality’ do not loop back to affect the initial two sets of drivers (anthropogenic impacts and natural hazards). Instead, this is an example of a flow diagram, with distinct start and end positions.

Figure 2. I am not convinced that the existing figure will be as useful to policy makers as claimed by the authors in the paper. At present the layout is confusing and rather difficult to digest. I believe it could be improved with some rethinking. I suggest at least

the following:

Use ‘Anthropogenic Drivers’ and ‘Natural Hazard Drivers’ as column headings at the top of the figure. Keep the three important outcomes (rice yield, rice growing area and rice quality) in a separate final row at the bottom of the figure.

Figure 2. Several other points:

Typhoon wind speed affects storm surge. The direct link is missing. Does flooding refer to river (freshwater) flooding or sea (saltwater) flooding? These need to be separated out somehow as they can both have major but different consequences, positive (e.g. fertile silt input, salt flushing) or negative (killing of standing crops, salt contamination of soil). Saltwater flooding needs to be linked to saltwater intrusion. Drought affects salt intrusion. The link is missing. Drought affects rice quality directly. The link is missing. Doesn’t the flow diagram need an ‘erosion’ box similar to Figure 3? Natural hazards such as typhoons and anthropogenic impacts (e.g. sediment starvation mentioned in the paper) will have consequences for both coastal erosion and river channel erosion. This needs further clarity.

Figure 3. Again, as with Fig.2, this is not strictly speaking a causal loop diagram, because ‘rice yield’ and ‘rice growing area’ do not loop back to the head of the figure to affect ‘climate change’. This is a flow diagram.

Figure 3. To improve clarity, keep the outcomes of rice yield and rice growing area in a separate row at the bottom of the figure.

Figure 2. Several other points:

Typhoon wind speed affects storm surge height. The direct link is missing. Surely pests and disease affect rice yield? The direct link is missing. Does the ‘erosion’ box refer to coastal erosion (shoreline retreat) or river channel erosion? This needs further consideration.

---

## Referee Comment (RC2) · Anonymous Referee #2 · 9 Oct 2020

Overall, this is a very well written paper, with only a few usage errors (missing some hyphens). This type of analysis is relevant for understanding rapid changes in the important delta regions producing significant proportions of food globally. This systems-level analysis is interesting and provides a structured way to understand the complex implications of climate change and human-cause land use changes on rice production in the Mekong River Delta and the Red River Delta.

However, given that both climate change- and human-induced changes are the objective of the study, there are some glaring omissions. The authors should consider how urban and exurban expansion, especially in the RRD, are decreasing rice yields.

As well as the shifting local and international markets for more diverse agricultural products (orchards, aquaculture) are changing rice distributions, and thus where the climate and human-cause impacts will persist. These are important land use changes that have downstream, so to speak, causations, particularly on available land for rice yields. The authors should seriously consider justifying why these variables where not included and also add to the discussion how they could be included, if the decision is to not include them in a revised version of the analysis.

---

## Author Comment (AC1) · 19 Nov 2020

We thank Reviewer 1 for taking the time to critically evaluate our manuscript and for providing constructive comments to improve our paper. Below, we reproduce all the comments that we have received from the reviewers and detail how we will address them in the revised version of our manuscript. Reference details for the citations used in our responses are provided as a list at the end of this document.

General comments

My overall general criticisms are: 1. That more information/data/evidence is needed

in support of the factors and processes that might impact rice cultivation on the two deltas investigated. At present there is mention of salt intrusion, erosion, sediment extraction, deltaic subsidence, fluvial sediment capture by dams, contamination, loss of soil quality, sea-level rise and increasing frequency/intensity of hazards. All of these are no doubt important for the long term sustainability of the MRD and RRD. But at the moment various statements by the authors about the relative importance of these processes need better grounding. There needs to be inclusion of published data where available, in order to help firm up the main arguments in the discussion. I fully understand that the overarching aim of the paper is to enable visualisation on how all the various influences are connected. But additional data from the literature are nonetheless still needed for proper substantiation.

2. Many minor grammatical errors need correcting throughout the manuscript.

3. Figures 2 and 3 need rethinking and should be improved.

The case studies in support of the factors and processes that might impact rice cultivation in the two deltas were provided in the supplementary materials. However, we agree with the Reviewer that we should include this information in the manuscript to support our arguments. We will work on including some of the case studies listed in the supplementary materials in the main text. In addition, we will fix the minor grammatical errors throughout the manuscript. Lastly, we agree that Figures 2 and 3 can be improved. We have worked on improving them based on the Reviewer's specific comments.

Specific comments

Specific comments and recommendations are listed as below. These are in order as they appear in the manuscript. I trust that the authors will not feel these are unnecessarily critical, but are offered in the spirit of improving the paper for eventual publication.

ABSTRACT L10. The authors use the term 'mega-deltas'. What does mega-delta

mean? This term needs to be defined early on somewhere in the body of the paper.

A mega-delta is a large, low-lying sedimentary landform located at the mouths of rivers. The mixing of fresh and saltwater in these sediment-rich land-ocean coastal zones provides fertile land for agricultural activities to support a large number of people. Besides agriculture, resources in mega-deltas have also been tapped for fisheries, navigation, trade, forestry, fossil energy production and manufacturing. Unfortunately, mega-deltas are highly vulnerable to a range of environmental hazards such as typhoons, floods, storm surges, tsunamis, coastal erosion and seasonal inundations. In addition, local human activities, land subsidence, water stresses and global sea level rise have exacerbated its environmental vulnerability (Day et al; 2016; Seto, 2011; Tessler et al. 2015). This information will be added to our manuscript.

L12. Remove 'happening'.

We will remove 'happening'.

L20. Change 'development' to 'growth' to avoid repeating 'development' in the sentence. Development will be changed to 'growth'. L21. Use a comma before 'which' or change to 'that'. This error needs correcting throughout the entire manuscript.

Comma will be added before "which".

L24. Hyphenate 'systems thinking' because this is used as an adjectival phrase, i.e. say 'a systems-thinking approach'. Do this similarly throughout the manuscript.

The phrase 'systems thinking' will be hyphenated throughout the manuscript.

INTRODUCTION

L44. Some references are needed here. The following paper and book chapter might be helpful: 'The 'terrific Tongking typhoon' of October 1881 – implications for the Red River Delta (northern Vietnam) in modern times.' Weather, 2012, 67: 72–75. [This gives an example of the severe effect of a typhoon on the rice harvest on the RRD.]

‘Impacts of climate change. Challenges of flooding in coastal East Asia.’ In: The Routledge Handbook of Environment and Society in Asia. P.G. Harris and G. Lang (Eds), Routledge, Oxford, pp.367–383. Chan, F.K.S., et al. 2015. [Mentions problems of flooding on the RRD and MRD, as well as other deltas in Asia.]

Thank you for suggesting these references. We will include them in our text. We will also use Terry et al. (2012) to highlight the dangers posed by a powerful typhoon with strong storm surge and the possibility of such an event happening in future in the Red River Delta.

L56. A reference is needed here in support of this statement.

We will include a reference by Normile (2013).

L59. Remove ‘related development’ (unnecessary text).

‘Related development’ will be removed.

L59. Remove ‘of’.

‘Of’ will be removed. L59-62. How can ‘coastal dikes. . .lead to a reduction in sediment and water availability. . .’? This misleading sentence needs rewriting.

Sentence will be changed to: ‘Infrastructure such as dikes reduces the availability of fertile silt for maintaining soil fertility. In addition, upstream dams, sand mining and ground water extraction have impacts on rice growing areas (Kondolf et al., 2018). Upstream dams and sand mining lead to a reduction in sediment availability and this may cause river channel incision and bank erosion as the sediment-deprived river water tend to erode the channel beds and banks (Kondolf, 1997). Conversely, groundwater extraction exacerbates land subsidence which coupled with rising sea levels may lead to a loss of agricultural land along the coasts (Allison et al., 2017; Robert, 2017; Schmitt et al., 2017).’

L77. Is the mention of ‘wicked’ problems helpful? What does this even mean? This

either needs explaining or omitting.

We use the term ‘wicked problems’ to refer to problems that have no clear definitions and have no easily identifiable, predefined solutions. They tend to come about due to a complexity and interdependency of components which create feedbacks and nonlinear responses to management interventions. The environmental problems present in the mega-deltas of Vietnam are caused by a range of interdependent anthropogenic and natural hazards drivers and the solutions to these problems are not straightforward. Hence, the use of systems-thinking is appropriate as analysis that considers divergent drivers and environmental effects can avoid oversimplifying a problem (Rittel and Webber, 1973; DeFries and Nagendra, 2017). This information will be added to our manuscript.

METHODS

L96. If the area of the MRD is 4 million ha, then how can the area of rice planted on the delta be 4.2 million ha? This is not possible, unless the authors are adding together the areas of subsequent plantings during the year. Please explain or correct.

The area of the Mekong Delta in Vietnam is 39,000 km2, equivalent to 3,900,000 ha, or 4 million ha (Schneider and Asch, 2020). In 2018, 4,107,200 ha of rice were planted in the Mekong River Delta with 24,506,900 tons of rice produced. The 4.2 million ha was derived from summing up the total planted areas for spring, autumn and winter paddies. The planted area for spring, autumn and winter paddies were 1,573,500 ha, 2,336,500 ha and 197,200 ha respectively (General Statistics Office of Vietnam, 2020). To avoid confusion, we will list the planted areas for each season in our revision.

L101. Is the RRD ‘floodplain’ area the same as the delta area? Please give the delta area to be consistent with the description of the MRD above.

The size of the Red River Delta in Vietnam is 15,000 km2, equivalent to 1.5 million ha (Schneider and Asch, 2020).

L107. Thick 'Quaternary accumulation' of what? Sand, silt or clay? Please briefly give more information on the character of the deltaic sediments.

Soils in the Red River Delta (RRD) consist of thick quaternary accumulation with loose and alternating sediment beds which are mostly organic in nature. In general, the Quaternary is divided into two sequences: the upper part consists of fine sediment clay, sandy clay and fine sand; and the lower part contains gravel with cobbles and coarse sand. The Quarternary sediments are underlain by a >400 m thick layer of Neogene sedimentary rocks that are made up of conglomerate sandstone, clay and siltstone (Berg et al., 2007). The additional information will be included in our revision.

L109. Remove 'slight'.

'Slight' will be removed

L110. Be more careful with grammar. The text should read '. . .the MRD has [not have]. . .while the RRD has [not have] a temperature of. . .' This type of error crops up many times throughout the manuscript, e.g. L118, L124, L124. Please ask a native English speaker to check the manuscript carefully for corrections.

Grammar will be corrected to 'has.'

L112. Request including climographs for the two deltas, so the reader can more easily understand the annual climatic cycles. Then mark on the graphs the planting, growing and harvesting times of the different rice crops.

We will include a climograph for the two deltas with the rice growing seasons and the different phrases of planting annotated in the revised manuscript.

L118. Please correct grammar.

'Has' will be changed to 'have'

L223. Please correct grammar.

‘Was’ will be changed to ‘were.’

L124. Please correct grammar.

‘Was’ will be changed to ‘were.’

L134. Change ‘. . .a major rice producing region’ to ‘major rice producing regions’.

‘A major rice producing region” will be changed to ‘major rice producing regions.’

L151. Change ‘describe’ to ‘describes’.

‘Describe’ will be changed to ‘describes.’

RESULTS

L193. The authors make a sweeping statement but without supporting data. What is the ‘extent of saltwater intrusion’? Is it possible to include a map here to show how saltwater intrusion has been extending into the delta over time?

In the Mekong Delta, 2.1 million ha of land is affected by salinity during the dry season (Pham et al., 2018). A map of the incidence and severity of salinity intrusion in the Mekong Delta from Preston et al. (2003) will be referenced.

L196. Where does the arsenic contamination come from? Anthropogenic sources? The authors need to explain otherwise the readers are left guessing.

The source of this arsenic contamination is from the groundwater. In the Mekong Delta, naturally occurring biochemical and hydrological processes cause As to be released from Fe oxides in rocks and sediments into groundwater reservoirs (Fendorf et al., 2010). In addition, deep groundwater extraction causes interbedded clays to compact and expel water containing dissolved As or As mobilizing solutes which are transferred into deep aquifer (Erban et al., 2013). Likewise, groundwater in the Red River Delta is also contaminated with high levels of As due to reductive dissolution of As from iron oxyhydroxides in buried sediment (Berg et al., 2007; Luu, 2019).

Even though farmers use river water instead of groundwater to irrigate their rice fields, the use of groundwater for other purposes and their eventual discharge into surrounding soils and rivers mean that soils and river water can become contaminated with As. Crop quality is reduced when the As enriched groundwater is deposited on topsoils and absorbed by plants during growth.

L203. How much sand mining is occurring? Is it for the construction of dikes? Again, some supporting data are needed.

Jordan et al (2019) compiled reported statistics of sand mining activities for the whole of the Vietnamese Mekong Delta and estimated that 177.77 Mm3 of sand was extracted in 2018. This value is likely an under-estimate as only reported amounts from local dredging contractors were available. Most of the sand was extracted for the local construction industry while only a small amount was used to maintain the river's channels.

L204. Again, another sweeping statement about the 'substantial reduction in sediment', but without any supporting data. Please substantiate better.

To support this statement about the 'substantial reduction in sediment', we refer to Park et al. (2020)'s study on the impact of extensive riverine mining on flood frequency in the Long Xuyen Quadrangle (LXQ) in the Mekong Delta. The LXQ had one of the highest sand extraction rates in Vietnam and there has been a significant decrease in flood frequency over the past 20 years, from 1995-2015. Daily water level series at local gauge stations showed an overall decreasing trend indicating that the lowering of the riverbed has reduced the frequency of flooding. The lack of significant changes in river discharge at a gauging station in Cambodia indicates that the lowering of the riverbed caused by sand mining in the Mekong has affected flood frequency trends more than any other climatic factors.

L204. How much 'land subsidence' has occurred, and over what period? Give rates if available. With all of the above, if the authors wish to include saltwater intrusion, sand

mining, sediment reduction and land subsidence in their Results section, then some additional supporting data are needed.

Based on time series data from 79 nested monitoring wells at 18 locations in the Mekong River Delta (MRD), Erban et al. (2014) found that compaction rate of sedimentary layers in the MRD is about 16 mm/year, similar to the 10-20 mm/year rates reported by Minderhoud et al. (2018). Likewise, Minderhoud et al. (2017) developed a 3D numerical groundwater flow model of the delta surface and concluded that subsidence rates from groundwater extraction is between 11 and 25 mm/year. The model also showed that 25 years of groundwater extraction since 1991 has resulted in a cumulative average of 18 cm of subsidence with hotspots recording over 30 cm of subsidence. Land subsidence from excessive groundwater extraction acts as a catalyst that increases vulnerability to saltwater intrusion and reduces the availability of land suitable for rice production.

Additional supporting information for saltwater intrusion, sand mining and sediment reduction will be added in the revised manuscript.

L225. This statement is wrong. Thermal expansion of seawater does not accelerate the melting of icecaps! Needs rewriting.

We will remove this sentence.

L234. The authors mention that the coastline is eroding at a rate of 5 to 10 mm/year. This rate seems far too low. A 0.5 cm rate of coastal retreat per year (0.5 m per century) is insignificant and suggests that the RRD and MRD have nothing to worry about. For comparison, Thailand's Chao Phraya delta front has experienced several km of shoreline retreat over recent decades. Please check the data.

Thank you for spotting this. It should be 5 to 10 m/year and not 5 to 10 mm/year.

L244. Change text to '…are affected by…' (similar use of plural also needed elsewhere).

‘Is affected by’ will be changed to ‘are affected by’

L245. What is the local rate of SLR along the coast of Vietnam or in the wider western South China Sea?

For Vietnam, observations at tide gauges show an average increase of 3.3 mm per year during the period 1993 to 2014 (Hens et al., 2018).

L246. Please give some information on the groundwater salinity thresholds for rice cultivation.

Rice is unable to thrive in the soil and water that have a salinity threshold of more than 4 g/L (Pham et al., 2018).

L258. Please correct grammar. Look out for similar errors elsewhere through the document.

‘Shift’ will be changed to ‘shifts.’

L266. Please rephrase this sentence. We change the sentence to ‘Drought-prone regions may experience longer and more frequent droughts in future.’

L272. This contradicts what was said in L118. In other words, what about the irrigation canals mentioned earlier in the paper? Aren’t these used for rice irrigation in the absence of sufficient rainfall?

Sentence will be deleted.

L273. Please correct grammar. Look out for similar errors elsewhere through the document.

“Exacerbates” will be changed to “exacerbate.”

L280. What is Cyrtorhinus? An insect, snake, bird, mammal? Please give the English name of this predator.

Cyrtorhinus lividipennis Reuter (Green Murid Bug) is an insect which predates on common rice pests such as planthoppers and leafhoppers.

DISCUSSION

L301. Please correct grammar.

‘Are’ will be changed to ‘is.’

L302. Please correct grammar.

‘Is’ will be changed to ‘are.’

L307. The authors have not provided any information on typhoon and drought frequencies experienced on the two deltas. Please give supporting information earlier in the paper.

An average of five to six typhoons affects Vietnam between June and November every year (Larson et al., 2014; Nguyen et al., 2007). Typhoon activity shifts from the north to the South as the year progresses. Therefore, peak activity in the north, central and southern part of Vietnam is in August, October and November respectively (Imamura and Dang, 1997). We reviewed the Digital Typhoon Database and found 303 typhoons that came within 500 km of Vietnam’s coastline from 1995 to 2018. 29 cyclones made its initial landfall in the Red River Delta while only four cyclones made landfall in the Mekong Delta during the study period – one each in 1973, 1996, 1997 and 2006. A total of 68 cyclones damaged rice crops during the study period (Unpublished results).

Vietnam was affected by droughts in 1997-1998, 2002-2003, 2009-2010 and most recently in 2015-2016. The 2015-2016 drought was the most severe in 90 years (Grosjean et al., 2016). All thirteen provinces in the Mekong Delta were affected by the 2015 drought. Compared to the Mekong, there is not much research or reports on droughts in the Red River Delta. The UNW-DPC (2014) reported that the Red River Delta experienced droughts from the end of 1998 to April 1999 which affected 86,140 ha of rice. Another drought occurred from January to February 2004 with the water level of the Red River at the lowest in 40 years. Low water levels were also reported in 2010,

however drought conditions and saltwater intrusion was more severe in the Mekong (Overland, 2010).

L313. '. . .high arsenic concentrations. . .likely due to geogenic conditions'. Please elaborate.

High arsenic concentrations in groundwater seem to be of natural origin. In the Mekong Delta, naturally occurring biochemical and hydrological processes cause As to be released from Fe oxides in rocks and sediments into groundwater reservoirs (Fendorf et al., 2010). In addition, deep groundwater extraction causes interbedded clays to compact and expel water containing dissolved As or As mobilizing solutes which are transferred into deep aquifer (Erban et al., 2013). Similarly, groundwater in the Red River Delta is also contaminated with high levels of As due to reductive dissolution of As from iron oxyhydroxides in buried sediment (Berg et al., 2007; Luu, 2019).

L327. Please correct grammar.

"Mean" will be changed to 'means.'

L338. Please correct grammar.

'Is' will be changed to 'are.'

L353. Please correct grammar.

'Is' will be changed to 'are.'

L356. Change 'practice' [noun] to 'practise' [verb].

'Practice' will be changed to 'practise.'

L364. Please correct grammar.

'Helps' will be changed to 'help.'

L371. Please correct grammar.

‘Benefits’ will be changed to ‘benefit.’

L386. Please correct grammar.

‘Is’ will be changed to ‘are.’

L386. Please correct grammar (later in the sentence). ‘Is’ will be changed to ‘are.’

CONCLUSIONS

L391. Use ‘Conclusions’.

‘Conclusion’ will be changed to ‘conclusions.’

L411. Please correct grammar.

The sentence will be changed to: ‘While the effects of climate change on food productivity are still uncertain. . .’

L414. Needs rephrasing. Do you mean ‘supporting large populations’? [the deltas support large populations – the populations do not support the deltas].

Across the world, deltas are global food production hubs that support large populations.

REFERENCES

L521. Reference is in the wrong place.

Thank you for noticing this. We will fix it.

FIGURES

Figure 1. A cardinal sin has been committed with the maps! Never use a word scale. ‘1 cm = 58 km’ will be incorrect if the published version of the map is not exactly the same size as the original. In fact, the scales must be wrong already as the three maps shown cannot all have the same scale. Use a scale bar instead on each map. These will be correct whatever size the maps are viewed or printed.

Noted. A scale bar will be used.

Figure 1. Increase the size of the long/lat text. Too small to read.

Font has been increased.

Figure 1. Label the countries.

Countries have been labelled. The changes to Figure 1 are reflected in the figure below.

Figure 1. L760-764. Unnecessary repetition. The long list of provinces is not needed, as they are already shown on the map.

Noted. We will remove the list of provinces.

Figure 2. Strictly speaking, it is incorrect to call this a causal loop diagram as claimed, because 'rice yield', 'rice growing area' and 'rice quality' do not loop back to affect the initial two sets of drivers (anthropogenic impacts and natural hazards). Instead, this is an example of a flow diagram, with distinct start and end positions.

We will call this a flow diagram.

Figure 2. I am not convinced that the existing figure will be as useful to policy makers as claimed by the authors in the paper. At present the layout is confusing and rather difficult to digest. I believe it could be improved with some rethinking. I suggest at least the following: Use 'Anthropogenic Drivers' and 'Natural Hazard Drivers' as column headings at the top of the figure. Keep the three important outcomes (rice yield, rice growing area and rice quality) in a separate final row at the bottom of the figure.

Figure 2. Several other points: Typhoon wind speed affects storm surge. The direct link is missing. Does flooding refer to river (freshwater) flooding or sea (saltwater) flooding? These need to be separated out somehow as they can both have major but different consequences, positive (e.g. fertile silt input, salt flushing) or negative (killing of standing crops, salt contamination of soil). Saltwater flooding needs to be linked to

saltwater intrusion. Drought affects salt intrusion. The link is missing. Drought affects rice quality directly. The link is missing. Doesn't the flow diagram need an 'erosion' box similar to Figure 3? Natural hazards such as typhoons and anthropogenic impacts (e.g. sediment starvation mentioned in the paper) will have consequences for both coastal erosion and river channel erosion. This needs further clarity.

We have taken the Reviewer's suggestion into consideration and reworked Figure 2. We separate anthropogenic and natural hazard drivers to reduce clutter. As such, typhoons and droughts were removed. In addition, we have added urban expansion, aquaculture expansion, fruit and vegetable production and agricultural intensification as these drivers would have a major impact on rice production. Lastly, we colour code each driver to represent whether the driver operates at a local or regional scale. A revamped Figure 2 is shown below.

Figure 3. Again, as with Fig.2, this is not strictly speaking a causal loop diagram, because 'rice yield' and 'rice growing area' do not loop back to the head of the figure to affect 'climate change'. This is a flow diagram.

Figure 3. To improve clarity, keep the outcomes of rice yield and rice growing area in a separate row at the bottom of the figure.

Figure 3. Several other points: Typhoon wind speed affects storm surge height. The direct link is missing. Surely pests and disease affect rice yield? The direct link is missing. Does the 'erosion' box refer to coastal erosion (shoreline retreat) or river channel erosion? This needs further consideration.

We have taken the reviewer's suggestions into consideration and edited Figure 3. We have colour coded each driver to represent whether the driver operates at a local or global scale. A revamped Figure 3 is shown below.

References used in response to reviewer's comments

[revised manuscript text omitted]

---

## Author Comment (AC2) · 19 Nov 2020

We thank the Reviewer 2 for taking the time to critically evaluate our manuscript and for providing constructive comments to improve our paper. Below, we reproduce all the comments that we have received from the reviewers and detail how we will address them in the revised version of our manuscript.

General comments

Overall, this is a very well written paper, with only a few usage errors (missing some hyphens). This type of analysis is relevant for understanding rapid changes in the

important delta regions producing significant proportions of food globally. This systems level analysis is interesting and provides a structured way to understand the complex implications of climate change and human-cause land use changes on rice production in the Mekong River Delta and the Red River Delta. However, given that both climate change- and human-induced changes are the objective of the study, there are some glaring omissions. The authors should consider how urban and exurban expansion, especially in the RRD, are decreasing rice yields.

As well as the shifting local and international markets for more diverse agricultural products (orchards, aquaculture) are changing rice distributions, and thus where the climate and human-cause impacts will persist. These are important land use changes that have downstream, so to speak, causations, particularly on available land for rice yields. The authors should seriously consider justifying why these variables where not included and also add to the discussion how they could be included, if the decision is to not include them in a revised version of the analysis.

Response

We are pleased that the Reviewer has recognized the value of using systems-thinking and flow diagrams to identify and visualize the interconnections among the drivers of rice productivity in both deltas. We agree with the comments made by the Reviewer that we have not taken into consideration the impacts of urbanization. We also agree that increased demand for more diverse agricultural products as well as environmental challenges such as saltwater intrusion has meant that farmers may be incentivized to convert existing rice cultivation areas into orchards, vegetable plots or aquaculture ponds. We have taken these factors into consideration and revised Figure 2 accordingly to reflect the importance of these anthropogenic drivers. The new information will also be added to our manuscript.

[Figure]

**Fig. 1.** Flow diagram showing the key anthropogenic drivers that affect rice production in the two mega-deltas of Vietnam.

---

## Author Response (AR1)

NANYANG TECHNOLOGICAL UNIVERSITY SINGAPORE

Asian School of the Environment
Nanyang Technological University of Singapore
50 Nanyang Avenue, Block N2-01C-43, Singapore 639798

Feb 20, 2021

**Editorial Board**

*Natural Hazards and Earth System Sciences*

Dear Dr Animesh Gain,

We thank the reviewers for their comments on our manuscript, "Interacting effects of land-use change and natural hazards on rice agriculture in the Mekong and Red River Deltas in Vietnam" (NHESS-2020-196). Recognizing this is a difficult time for all of us, we are grateful to the referees for committing their time to this review.

We have revised our manuscript based on the reviewers' comments. We have tried to address all comments as best possible and all suggestions have been taken into consideration in our revisions.

We look forward to hearing back from the editorial team. Thank you for your consideration.

On behalf of our co-authors,
Yuen Kai Wan &
Janice Ser Huay Lee

Asian School of the Environment
Nanyang Technological University of Singapore
50 Nanyang Avenue, Block N2-01C-43, Singapore 639798

**Handling Reviewers' comments:**

**Reviewer #1:**

**General comments**

**1. My overall general criticisms are:**
1. That more information/data/evidence is needed in support of the factors and processes that might impact rice cultivation on the two deltas investigated. At present there is mention of salt intrusion, erosion, sediment extraction, deltaic subsidence, fluvial sediment capture by dams, contamination, loss of soil quality, sea-level rise and increasing frequency/intensity of hazards. All of these are no doubt important for the long term sustainability of the MRD and RRD. But at the moment various statements by the authors about the relative importance of these processes need better grounding. There needs to be inclusion of published data where available, in order to help firm up the main arguments in the discussion. I fully understand that the overarching aim of the paper is to enable visualisation on how all the various influences are connected. But additional data from the literature are nonetheless still needed for proper substantiation.

2. Many minor grammatical errors need correcting throughout the manuscript.

3. Figures 2 and 3 need rethinking and should be improved.

**Response: We thank the reviewer for this response and agree. We included more information to support the factors and processes that might impact rice cultivation in the two deltas. Instead of summarizing everything in one section, we created additional headings such as 'local anthropogenic drivers', 'regional anthropogenic drivers', 'local natural hazard drivers' and 'global natural hazard driver' to highlight the nature of the driver and the scale at which each factor operates. For 'local anthropogenic drivers', subheadings such as 'aquaculture and alternative crops', 'urban expansion', 'dikes', 'sand mining' and 'groundwater extraction' were created to inform the reader which factor we focused on. Sub-headings were also created for 'regional anthropogenic drivers', 'local natural hazard drivers' and 'global natural hazard driver.' This approach allowed us to describe the factors and processes in-depth and include appropriate case studies to support our arguments.**

**The grammatical errors throughout the manuscript have been corrected. We have improved the two flow diagrams (previously known as Figures 2 and 3). We hope the revised figure is neater and clearer for the reader.**

**Specific comments**

Specific comments and recommendations are listed as below. These are in order as they appear in the manuscript. I trust that the authors will not feel these are unnecessarily critical, but are offered in the spirit of improving the paper for eventual publication.

**ABSTRACT**
**2. L10. The authors use the term 'mega-deltas'. What does mega-delta mean? This term needs to be defined early on somewhere in the body of the paper.**

**Response: For simplicity, we decided to use the term 'delta' instead of 'mega-delta.' On the reviewer's suggestion, we have defined 'delta' in the introduction at the beginning of our manuscript. Please see the following:**

**(Page 2, line 40) A delta is defined as a low-lying sedimentary landform located at the mouths of rivers. The mixing of fresh and saltwater in these sediment-rich land-ocean coastal zones provides fertile land for agricultural activities to support a large number of people. Besides agriculture, resources in deltas have also been tapped for fisheries, navigation, trade, forestry, fossil energy production and manufacturing. Unfortunately, deltas are highly vulnerable to a range of environmental hazards such as typhoons, floods, storm surges, tsunamis, coastal erosion and seasonal inundations (Syvitski and Saito, 2007). In addition,**

**local human activities, land subsidence, water stresses and global sea level rise have exacerbated their environmental vulnerability (Day et al; 2016; Seto, 2011; Tessler et al. 2015).**

**3. L12. Remove 'happening'.**

**Response: The sentence has been removed from the manuscript as we have re-written parts of the abstract.**

**4. L20. Change 'development' to 'growth' to avoid repeating 'development' in the sentence.**

**Response: We agree. 'Economic development' was changed to 'economic growth' to avoid repetition. Please see the following:**

**(Page 1, line 21) Notably, anthropogenic developments meant to improve agricultural productivity or increase economic growth can create many unwanted environmental consequences such as an increase in flooding, saltwater intrusion and land subsidence, which in turn decreases rice production and quality.**

**5. L21. Use a comma before 'which' or change to 'that'. This error needs correcting throughout the entire manuscript.**

**Response: We agree. A comma was added before "which". Please see the following:**

**(Page 1, line 21) Notably, anthropogenic developments meant to improve agricultural productivity or increase economic growth can create many unwanted environmental consequences such as an increase in flooding, saltwater intrusion and land subsidence, which in turn decreases rice production and quality.**

**6. L24. Hyphenate 'systems thinking' because this is used as an adjectival phrase, i.e. say 'a systems-thinking approach'. Do this similarly throughout the manuscript.**

**Response: We thank the reviewer for this comment and agree. The phrase 'systems thinking' has been hyphenated throughout the manuscript, i.e., 'systems-thinking.' Please see lines 14, 25, 28.**

**INTRODUCTION**

**7. L44. Some references are needed here. The following paper and book chapter might be helpful:**
'The 'terrific Tongking typhoon' of October 1881 – implications for the Red River Delta (northern Vietnam) in modern times.' Weather, 2012, 67: 72–75. [This gives an example of the severe effect of a typhoon on the rice harvest on the RRD.]
'Impacts of climate change. Challenges of flooding in coastal East Asia.' In: The Routledge Handbook of Environment and Society in Asia. P.G. Harris and G. Lang (Eds), Routledge, Oxford, pp.367–383. Chan, F.K.S., et al. 2015. [Mentions problems of flooding on the RRD and MRD, as well as other deltas in Asia.]

**Response: We thank the reviewer for these references. We have included these two references in our manuscript to highlight the dangers posed by typhoons and flooding. See the following:**

**(Page 2, line 74) Besides land-use change, rice grown in the RRD and MRD are susceptible to damage from natural hazards such as typhoons, floods and droughts (Chan et al., 2012; 2015; Grosjean et al., 2016; Terry et al., 2012).**

**In addition, we have used Terry et al. (2012) to highlight the dangers posed by a powerful typhoon with a strong storm surge and the possibility of such an event happening in future. See the following:**

**(Page 13, line 560) Besides sea level rise, climate change may also exacerbate the effects of natural hazards by increasing the frequency and severity of natural disasters (cf. Hausfather et al., 2017; Grosjean et al., 2016; Terry et al., 2012).**

**8. L56. A reference is needed here in support of this statement.**

**Response: We moved the sentence to the results section and included a reference by Normile (2013). See the following:**

**(Page 6, line 242) For example, killing plant hoppers now requires a pesticide dose 500 times more than was needed in the past (Normile, 2013).**

**9. L59. Remove ‘related development’ (unnecessary text).**

**Response: We thank the reviewer for spotting this. We have removed the sentence as we have re-written the introduction.**

**10. L59. Remove ‘of’.**

**Response: We thank the reviewer for spotting this. We have removed the sentence as we have re-written the introduction.**

**11. L59-62. How can ‘coastal dikes…lead to a reduction in sediment and water availability…’? This misleading sentence needs rewriting.**

**Response: We thank the reviewer for this comment. We have removed the sentence as we have re-written the introduction.**

**12. L77. Is the mention of ‘wicked’ problems helpful? What does this even mean? This either needs explaining or omitting.**

**Response: We thank the reviewer for this comment and agree. We believe it is helpful and we use the term ‘wicked problems’ to refer to problems that have no clear definitions and have no easily identifiable, predefined solutions. Traced to discussions on policy studies in the 1970s (cf. Rittel and Webber, 1973), the term ‘wicked problems’ came about due to an interdependency of components which create feedbacks and nonlinear responses to management interventions. The environmental problems present in the deltas of Vietnam can be considered “wicked” as they are caused by a range of interdependent anthropogenic and natural hazards drivers and the solutions to these problems are not straightforward (Rittel and Webber, 1973; DeFries and Nagendra, 2017). Please see the following:**

**(Page 3, line 95) The use of systems-thinking is appropriate as the “wicked” environmental problems present in the deltas of Vietnam are caused by a range of interdependent anthropogenic and natural hazards drivers operating at multiple scales with no easily identifiable, predefined solutions. While interventions may be made to ameliorate problems, these interventions may create feedbacks and unanticipated outcomes (Rittel and Webber, 1973; DeFries and Nagendra, 2017).**

METHODS

**13. L96. If the area of the MRD is 4 million ha, then how can the area of rice planted on the delta be 4.2 million ha? This is not possible, unless the authors are adding together the areas of subsequent plantings during the year. Please explain or correct.**

**Response: We thank the reviewer for this comment and agree. In our revision, we have listed the planted areas for each season to avoid confusion. Please see the following:**

**(Page 3, line 114) The Mekong River Delta (MRD) is the world's third largest delta with a physical area of 4 million ha and it is the larger of the two deltas in Vietnam (Schneider and Asch, 2020; Figure 1). In 2018, the planted area for spring, autumn and winter paddies was 1,573.5 thousand ha, 2,336.5 thousand ha and 197.2 thousand ha respectively. In total, 4.1 million ha of rice was planted over the three planting seasons with 24,507 thousand tons of rice produced.**

**14. L101. Is the RRD 'floodplain' area the same as the delta area? Please give the delta area to be consistent with the description of the MRD above.**

**Response: We thank the reviewer and agree with this comment. We removed the phrase 'floodplain area' and replaced it with 'delta area' to be consistent with the description of the MRD. Please see the following:**

**(Page 3, line 121) Up north, the Red River Delta (RRD) is the next largest with a physical delta area of 1.5 million ha (Figure 1; Schneider and Asch, 2020).**

**15. L107. Thick 'Quaternary accumulation' of what? Sand, silt or clay? Please briefly give more information on the character of the deltaic sediments.**

**Response: We thank the reviewer for this comment. We have provided more information on the character of the deltaic sediments for clarity. Please see the following:**

**(Page 3, line 128) Conversely, soils in the RRD consist of Holocene delta sediments. These Holocene delta sediments are relatively fine-grained muds and sands, up to 30 m thick that are the product of rapid progradation during the Holocene high sea level stand (Mathers and Zalasiewicz, 1999). The Holocene sequence overlies coarse-grained Pleistocene sediments dominated by braided river and alluvial fan deposits formed during the last glacial low sea level stand. The Quaternary sediments are underlain by a >400 m thick layer of Neogene sedimentary rocks that are made up of conglomerate sandstone, clay and siltstone (Berg et al., 2007).**

**16. L109. Remove 'slight'.**

**Response: We thank the reviewer for this comment. The sentence has been removed from the manuscript as we rewrote the paragraph.**

**17. L110. Be more careful with grammar. The text should read '…the MRD has [not have]…while the RRD has [not have] a temperature of…' This type of error crops up many times throughout the manuscript, e.g. L118, L124, L124. Please ask a native English speaker to check the manuscript carefully for corrections.**

**Response: We thank the reviewer for spotting these. We agree and have changed the wording where necessary. In this case, the sentence has been removed as we re-wrote the paragraph.**

Asian School of the Environment
Nanyang Technological University of Singapore
50 Nanyang Avenue, Block N2-01C-43, Singapore 639798

**18. L112. Request including climographs for the two deltas, so the reader can more easily understand the annual climatic cycles. Then mark on the graphs the planting, growing and harvesting times of the different rice crops.**

**Response: We thank the reviewer for this comment and agree. We have included a climograph for each of the two deltas with the rice growing seasons. The different phases of rice planting, growing and harvesting are represented by the colour gradient in the revised manuscript. Please see our revised figures on page 28:**

[Figure]

**Figure 2. Climograph for the Mekong River Delta annotated with the winter-spring, summer-autumn and autumn-winter growing seasons. The colour gradient (from faded to dark) represents planting, growing and harvesting times for each crop. Precipitation and temperature data were taken from the Statistical Yearbook of Vietnam 2018 (General Statistics Office, 2018).**

[Figure]

**Figure 3. Climograph for the Red River Delta annotated with the spring and autumn growing seasons. The colour gradient represents planting, growing and harvesting times for each crop. Precipitation and temperature data were taken from the Statistical Yearbook of Vietnam 2018 (General Statistics Office, 2018).**

**19. L118. Please correct grammar.**

**Response: We thank the reviewer for this comment. 'Has' was changed to 'have.' Please see the following:**

**(Page 4, line 147) In the MRD, favorable environmental conditions with ample rainfall, tropical temperatures and fertile alluvial soils, coupled with an extensive dike and irrigation system, have facilitated the production of three rice crops annually: winter-spring, summer-autumn and autumn-winter (Table 1; Figure 2).**

**20. L223. Please correct grammar.**

**Response: We thank the reviewer for this comment. 'Was' was changed to 'were.' Please see the following:**

**(Page 4, line 153) The chilly winters preclude the cultivation of a third crop of rice. Approximately 3,507 thousand tons of rice were produced during the spring cropping season while 2,789.1 thousand tons were produced during the autumn season in 2018 (General Statistics Office of Vietnam, 2020).**

**21. L124. Please correct grammar.**

**Response: We thank the reviewer for this comment. We changed this to 'were.' Please see the following:**

**(Page 4, line 154) Approximately 3,507 thousand tons of rice were produced during the spring cropping season while 2,789.1 thousand tons were produced during the autumn season in 2018 (General Statistics Office of Vietnam, 2020).**

**22. L134. Change '…a major rice producing region' to 'major rice producing regions'.**

**Response: We thank the reviewer for this comment. 'A major rice producing region" was changed to 'major rice producing regions.' Please see the following:**

**(Page 4, line 165) Since sea level rise would affect the viability of the two deltas as major rice producing regions (Mainuddin et al., 2006), we also included relevant articles on sea level rise**.

**23. L151. Change 'describe' to 'describes'.**

**Response: We thank the reviewer for this comment. 'Describe' was changed to 'describes.' Please see the following:**

**(Page 5, line 180) We constructed two flow diagrams - the first flow diagram describes how anthropogenic land-use drivers affect rice growing area, rice yield per hectare and rice quality in the MRD and RRD (Figure 4), while the second causal flow diagram describes how natural hazards in the MRD and RRD affect rice growing area, rice yield per hectare and rice quality (Figure 6).**

**RESULTS**

**24. L193. The authors make a sweeping statement but without supporting data. What is the 'extent of saltwater intrusion'? Is it possible to include a map here to show how saltwater intrusion has been extending into the delta over time?**

**Response: We thank the reviewer for this comment. We added information on the extent of saltwater intrusion but we are unable to include a map of how saltwater intrusion has been extending into the delta over time due to copyright issues. We include a reference by Preston et al. (2003) should the reader wish to see a map of saltwater intrusion in the Mekong Delta. Please see the following:**

**(Page 5, line 188) In the Mekong Delta, salinity intrusion is a naturally occurring phenomenon during the dry season. Tides from the South China Sea and the Gulf of Thailand bring saltwater inland and salinity intrudes up to 70-90 km inland as the length of sea dikes is limited. There are 1,500 km of sea and estuary dikes in RRD versus 450 km of sea dikes in the MRD (Le et al., 2018; Preston et al., 2003; Pilarcyzk and Nguyen, 2005).**

**25. L196. Where does the arsenic contamination come from? Anthropogenic sources? The authors need to explain otherwise the readers are left guessing.**

**Response: We thank the reviewer for this comment. The source of this arsenic contamination is groundwater. Even though farmers use river water instead of groundwater to irrigate their rice fields, the use of groundwater for other purposes and their eventual discharge into surrounding soils and rivers mean that soils and river water can become contaminated with As. Crop quality is reduced when the As enriched groundwater is deposited on topsoils and absorbed by plants during growth. Information related to the source of arsenic in groundwater was added to our manuscript. Please see the following:**

**(Page 8, line 324) High arsenic concentrations in groundwater seem to be of natural origin. In the Mekong Delta, naturally occurring biochemical and hydrological processes cause As to be released from Fe oxides in rocks and sediments into groundwater reservoirs (Fendorf et al., 2010). In addition, deep groundwater extraction causes interbedded clays to compact and expel water containing dissolved As (Erban et al., 2013). Crop quality is reduced when the arsenic enriched water is deposited on topsoils and absorbed by rice plants during growth (Rahman and Hasegawa, 2011; Figure 4).**

**(Page 8, line 332) Groundwater in the Red River Delta is also contaminated with high levels of As due to reductive dissolution of As from iron oxyhydroxides in buried sediment (Berg et al., 2007; Luu, 2019).**

**26. L203. How much sand mining is occurring? Is it for the construction of dikes? Again, some supporting data are needed.**

**Response: We thank the reviewer for this comment. We added details related to sand mining, please see the following:**

**(Page 7, line 282) Fueled by demand from reclamation, export and construction, 55.2 million tons of sediment were extracted from the Mekong main stem in Laos, Thailand, Cambodia and Vietnam from 2011 to 2012 (Bravard and Gaillot, 2013; Robert, 2017). A more recent analysis of bathymetric maps and the local refilling processes by Jordan et al. (2019) put the amount of sand extracted from the Mekong Delta in 2018 at 17.77 $Mm^3$.**

**27. L204. Again, another sweeping statement about the 'substantial reduction in sediment', but without any supporting data. Please substantiate better.**

**Response: We thank the reviewer for this comment. To support this statement about the 'substantial reduction in sediment', we refer to Park et al. (2020)'s study on the impact of extensive riverine mining on flood frequency in the Long Xuyen Quadrangle (LXQ) in the Mekong Delta. Please see the following:**

**(Page 7, line 295) A recent study of riverine mining on flood frequency in the Long Xuyen Quadrangle (LXQ) in the Mekong Delta found that flood frequency had dropped by 7.8% from 2005-2015. Water levels at local gauge stations also showed an overall decreasing trend indicating that the lowering of the riverbed has reduced the frequency of flooding.**

**28. L204. How much 'land subsidence' has occurred, and over what period? Give rates if available. With all of the above, if the authors wish to include saltwater intrusion, sand mining, sediment reduction and land subsidence in their Results section, then some additional supporting data are needed.**

**Response: We thank the reviewer for this comment. A case study on land subsidence in the Mekong delta has been included in our manuscript. Please see the following:**

Asian School of the Environment
Nanyang Technological University of Singapore
50 Nanyang Avenue, Block N2-01C-43, Singapore 639798

**(Page 7, line 315) Minderhoud et al. (2017) developed a 3D numerical groundwater flow model of the MRD surface and found that subsidence rates from groundwater extraction were between 1.1 and 2.5 cm/year. The model also showed that 25 years of groundwater extraction since 1991 has resulted in a cumulative average of 18 cm of subsidence with some hotspots recording over 30 cm of subsidence.**

**Additional supporting information for saltwater intrusion, sand mining and sediment reduction has been added in the revised manuscript (cf. results section).**

**29. L225. This statement is wrong. Thermal expansion of seawater does not accelerate the melting of icecaps! Needs rewriting.**

**Response: We thank the reviewer for spotting this. We have removed this erroneous sentence as well as the entire section on climate change. We decided to focus more on the natural hazards aspect and make references to the effects of climate change in the section on sea level rise as well as the discussion section.**

**30. L234. The authors mention that the coastline is eroding at a rate of 5 to 10 mm/year. This rate seems far too low. A 0.5 cm rate of coastal retreat per year (0.5 m per century) is insignificant and suggests that the RRD and MRD have nothing to worry about. For comparison, Thailand's Chao Phraya delta front has experienced several km of shoreline retreat over recent decades. Please check the data.**

**Response: We thank the reviewer for spotting this. We have removed this sentence as we have removed the entire section on climate change. We decided to focus more on the natural hazards aspect and make references to the effects of climate change in the section on sea level rise as well as the discussion section.**

**31. L244. Change text to '…are affected by…' (similar use of plural also needed elsewhere).**

**Response: We thank the reviewer for this comment. 'Is affected by' has been changed to 'are affected by.' Please see the following:**

**(Page 12, line 528) Approximately 1.8 million ha in the MRD are already affected by dry season salinity of which 1.3 million ha are affected by salinity levels above 5 g/L (Lassa et al., 2016).**

**32. L245. What is the local rate of SLR along the coast of Vietnam or in the wider western South China Sea?**

**Response: For Vietnam, observations at tide gauges show an average increase of 3.3 mm per year during the period 1993 to 2014 (Hens et al., 2018). Please see the following:**

**(Page 12, line 518) To quantify sea level rise locally, observations at tide gauges across Vietnam have recorded an average yearly increase of 3.3 mm from 1993-2014 (Hens et al., 2018).**

**33. L246. Please give some information on the groundwater salinity thresholds for rice cultivation.**

**Response: We thank the reviewer for this comment. Please see the following for additional information on groundwater salinity thresholds for rice cultivation:**

**(Page 5, line 193) As rice plants are unable to thrive in soils with soil salinities exceeding 4 g/L (Pham et al., 2018b), affected farmers have converted their paddy fields into aquaculture ponds to cultivate shrimp and fish instead.**

**34. L258. Please correct grammar. Look out for similar errors elsewhere through the document.**

**Response: We thank the reviewer for spotting this and have been more mindful of the grammar. In this case we have removed the sentence.**

**35. 'Shift' will be changed to 'shifts.'**

**Response: We thank the reviewer for this comment. We have removed the section on climate change to focus more on natural hazards.**

**36. L266. Please rephrase this sentence.**

**Response: We thank the reviewer for this comment. We have removed the section on climate change to focus more on natural hazards.**

**37. L272. This contradicts what was said in L118. In other words, what about the irrigation canals mentioned earlier in the paper? Aren't these used for rice irrigation in the absence of sufficient rainfall?**

**Response: We thank the reviewer for this comment. We have removed the section on climate change to focus more on natural hazards.**

**38. L273. Please correct grammar. Look out for similar errors elsewhere through the document.**

**Response: We thank the reviewer for this comment. We have removed the section on climate change to focus more on natural hazards.**

**39. L280. What is Cyrtorhinus? An insect, snake, bird, mammal? Please give the English name of this predator.**

**Response: We thank the reviewer for this comment. We have removed the section on climate change. In any case, *Cyrtorhinus lividipennis* Reuter (Green Murid Bug) is an insect which predates on common rice pests such as planthoppers and leafhoppers.**

**DISCUSSION**

**40. L301. Please correct grammar.**

**Response: We thank the reviewer for this comment. 'Are' was changed to 'is.' Please see the following:**

**(Page 13, line 564) The use of flow diagrams provides a visual overview of the key anthropogenic drivers and natural hazards that affect rice production but we caution that Red River Delta and the Mekong River Delta are vast and diverse regions and there are differences in the ways each delta is affected by natural hazards and anthropogenic drivers.**

**41. L302. Please correct grammar.**

**Response: We thank the reviewer for this comment. 'Is' was changed to 'are.' Please see the following:**

**(Page 13, line 567) For example, high dikes and the associated problem of sediment exclusion are a problem unique to the Mekong Delta (Chapman et al., 2017).**

**42. L307. The authors have not provided any information on typhoon and drought frequencies experienced on the two deltas. Please give supporting information earlier in the paper.**

**Response: We thank the reviewer for this comment. More information on typhoon and drought frequencies in the two deltas was furnished in the manuscript. Please see the following:**

**(Page 9, line 401) Vietnam was affected by droughts in 1997-1998, 2002-2003, 2009-2010 and most recently in 2015-2016. The 2015-2016 drought was the most severe in 90 years (Grosjean et al., 2016). All thirteen provinces in the Mekong Delta were affected by the 2015 drought.**

**(Page 10, line 410) The UNW-DPC (2014) reported that the RRD experienced droughts from the end of 1998 to April 1999 which affected 86,140 ha of rice. Another drought occurred from January to February 2004 with the water level of the Red River at the lowest in 40 years. Low water levels were also reported in 2010, however drought conditions and saltwater intrusion was more severe in the MRD (Overland, 2010).**

**(Page 11, line 468) An average of five to six typhoons affects Vietnam between June and November every year (Larson et al., 2014; Nguyen et al., 2007). Typhoon activity shifts from the north to the South as the year progresses. Therefore, peak activity in the north and southern part of Vietnam is in August and November respectively (Imamura and Dang, 1997). We reviewed the Digital Typhoon Database and found 303 typhoons that came within 500 km of Vietnam's coastline from 1995 to 2018. 29 cyclones made their initial landfall in the Red River Delta while only four cyclones made landfall in the Mekong Delta during the study period – one each in 1973, 1996, 1997 and 2006 (Unpublished results).**

**43. L313. '…high arsenic concentrations…likely due to geogenic conditions'. Please elaborate.**

**Response: We thank the reviewer for this comment. High arsenic concentrations in groundwater seem to be of natural origin. This was elaborated on in an earlier point on the origins of arsenic in ground (cf. comment #25). See the following:**

**(Page 8, line 324) High arsenic concentrations in groundwater seem to be of natural origin. In the Mekong Delta, naturally occurring biochemical and hydrological processes cause As to be released from Fe oxides in rocks and sediments into groundwater reservoirs (Fendorf et al., 2010). In addition, deep groundwater extraction causes interbedded clays to compact and expel water containing dissolved As (Erban et al., 2013). Crop quality is reduced when the arsenic enriched water is deposited on topsoils and absorbed by rice plants during growth (Rahman and Hasegawa, 2011; Figure 4).**

**(Page 8, line 332) Groundwater in the Red River Delta is also contaminated with high levels of As due to reductive dissolution of As from iron oxyhydroxides in buried sediment (Berg et al., 2007; Luu, 2019).**

**44. L327. Please correct grammar.**

**Response: We thank the reviewer for this comment. We have removed this sentence as we have re-written the parts of the 'untangling complexity' section to improve it.**

**45. L338. Please correct grammar.**

**Response: We thank the reviewer for this comment. We have removed this sentence as we have re-written parts of the 'untangling complexity' section to improve it.**

**46. L353. Please correct grammar.**

**Response: We thank the reviewer for spotting this. We have removed this sentence as we have re-written large parts of the section to point out that farmers do not passively accept their fate but attempt to adapt to their environments and improvise to come up with solutions to overcome difficulties. We have edited the section heading to 'adaptations and soft solutions' to better reflect the changes made.**

**47. L356. Change 'practice' [noun] to 'practise' [verb].**

**Response: We thank the reviewer for this comment. 'Practice' was changed to 'practise.' Please see the following:**

**(Page 14, line 618) Farmers who practise IPM use a combination of pest resistant cultivars, fertilizer management and agronomic practices to increase the effects of predators and other naturally occurring biological control agents.**

**48. L364. Please correct grammar.**

**Response: We thank the reviewer for this comment. 'Helps' was changed to 'help.' Please see the following:**

**(Page 14, line 627) Instead, fish help to control pests and fish droppings keep the soil fertile. Upon maturity, the fish can be sold to increase the farmer's income by up to 30% (Berg et al., 2017; Bosma et al., 2012).**

**49. L371. Please correct grammar.**

**Response: We thank the reviewer for this comment. 'Benefits' was changed to 'benefit.' Please see the following:**

**(Line 14, line 633) Overall, the higher incomes and ecosystem services provided by the fish or ducks, coupled with reduced agrochemical use benefit farmers.**

**50. L386. Please correct grammar.**

**Response: We thank the reviewer for spotting this. We have removed this sentence as we have re-written large parts of the section to point out that farmers do not passively accept their fate but attempt to adapt to their environments and improvise to come up with solutions to overcome difficulties. We have edited the section heading to 'adaptations and soft solutions' to better reflect the changes made.**

**51. L386. Please correct grammar (later in the sentence).**

**Response: We thank the reviewer for spotting this. We have removed this sentence as we have re-written large parts of the section to point out that farmers do not passively accept their fate but attempt to adapt to their environments and improvise to come up with solutions to overcome difficulties. We have edited the section heading to 'adaptations and soft solutions' to better reflect the changes made.**

**CONCLUSIONS**

**52. L391. Use 'Conclusions'.**

**Response: We thank the reviewer for this comment. 'Conclusion' has been changed to 'conclusions.'**

**53. L411. Please correct grammar.**

**Response: We thank the reviewer for this comment. We have removed this sentence as we re-wrote parts of the 'conclusions.'**

**54. L414. Needs rephrasing. Do you mean 'supporting large populations'? [the deltas support large populations – the populations do not support the deltas].**

**Response: We thank the reviewer for this comment. We have rephrased this sentence. Please see the following:**

**(Page 16, line 682) Across the world, deltas are global food production hubs that support large populations.**

Use 'Anthropogenic Drivers' and 'Natural Hazard Drivers' as column headings at the top of the figure. Keep the three important outcomes (rice yield, rice growing area and rice quality) in a separate final row at the bottom of the figure.

Figure 2. Several other points:
Typhoon wind speed affects storm surge. The direct link is missing. Does flooding refer to river (freshwater) flooding or sea (saltwater) flooding? These need to be separated out somehow as they can both have major but different consequences, positive (e.g. fertile silt input, salt flushing) or negative (killing of standing crops, salt contamination of soil). Saltwater flooding needs to be linked to saltwater intrusion. Drought affects salt intrusion. The link is missing. Drought affects rice quality directly. The link is missing. Doesn't the flow diagram need an 'erosion' box similar to Figure 3? Natural hazards such as typhoons and anthropogenic impacts (e.g. sediment starvation mentioned in the paper) will have consequences for both coastal erosion and river channel erosion. This needs further clarity.

**Response: We thank the reviewer for this comment and agree with the reviewer. We have taken Reviewer 1's suggestion into consideration and reworked Figure 2. In our new figure, we separate anthropogenic and natural hazard drivers to reduce clutter. As such, typhoons and droughts were removed. In addition, we have added urban expansion, aquaculture expansion, fruit and vegetable production and agricultural intensification as these drivers have a major impact on rice production. Lastly, we colour code each driver to represent whether the driver operates at a local or regional scale. A revised Figure 2 is shown here.**

[Figure]

**63. Figure 3. Again, as with Fig.2, this is not strictly speaking a causal loop diagram, because ‘rice yield’ and ‘rice growing area’ do not loop back to the head of the figure to affect ‘climate change’. This is a flow diagram.**

Figure 3. To improve clarity, keep the outcomes of rice yield and rice growing area in a separate row at the bottom of the figure.

Figure 3. Several other points:
Typhoon wind speed affects storm surge height. The direct link is missing. Surely pests and disease affect rice yield? The direct link is missing. Does the ‘erosion’ box refer to coastal erosion (shoreline retreat) or river channel erosion? This needs further consideration.

**Response: We thank the reviewer for this comment and agree with the reviewer. We have taken the reviewer’s suggestions into consideration and edited Figure 3. We have colour coded each driver to represent whether the driver operates at a local or global scale. A revised Figure 3 is shown here.**

[Figure]

**Reviewer #2:**

**General comments**

**1. Overall, this is a very well written paper, with only a few usage errors (missing some hyphens). This type of analysis is relevant for understanding rapid changes in the important delta regions producing significant proportions of food globally. This systems level analysis is interesting and provides a structured way to understand the complex implications of climate change and human-cause land use changes on rice production in the Mekong River Delta and the Red River Delta. However, given that both climate change- and human-induced changes are the objective of the study, there are some glaring omissions. The authors should consider how urban and exurban expansion, especially in the RRD, are decreasing rice yields.**

As well as the shifting local and international markets for more diverse agricultural products (orchards, aquaculture) are changing rice distributions, and thus where the climate and human-cause impacts will persist. These are important land use changes that have downstream, so to speak, causations, particularly on available land for rice yields. The authors should seriously consider justifying why these variables where not included and also add to the discussion how they could be included, if the decision is to not include them in a revised version of the analysis.

**Response: We thank Reviewer 2 for these kind words and are pleased that the reviewer recognizes the value of using systems-thinking and flow diagrams to identify and visualize the interconnections among the drivers of rice productivity in both deltas. We agree with the comments made by the Reviewer that we have not taken into consideration the impacts of urbanization. We also agree that increased demand for more diverse agricultural products as well as environmental challenges such as saltwater intrusion has meant that farmers may be incentivised to convert existing rice cultivation areas into orchards, vegetable plots or aquaculture ponds. We have taken these factors into consideration and revised Figure 2 accordingly to reflect the importance of these anthropogenic drivers. In the results section, we have added new sub-sections on 'aquaculture and alternative crops' and 'urban expansion" to elaborate on these factors. Please see the following:**

**(Page 5, line 187) Aquaculture and alternative crops**

[revised manuscript text omitted]